# Vivianite formation in ferruginous sediments from Lake Towuti, Indonesia.

Aurèle Vuillemin[1,2], André Friese[1], Richard Wirth[1], Jan A. Schuessler[1], Anja M. Schleicher[1], Helga Kemnitz[1], Andreas Lücke[3], Kohen W. Bauer[4,5], Sulung Nomosatryo[1,6], Friedhelm von Blanckenburg[1], Rachel Simister[4], Luis G. Ordoñez[7], Daniel Ariztegui[7], Cynthia Henny[6], James M. Russell[8], Satria Bijaksana[9], Hendrik Vogel[10], Sean A. Crowe[4,5,11], Jens Kallmeyer[1] and the Towuti Drilling Project Science Team*.

[1]GFZ German Research Centre for Geosciences, Helmholtz Centre Potsdam, Potsdam, 14473, Germany
[2]Present address: Department of Earth and Environmental Science, Paleontology and Geobiology, Ludwig-Maximilians-Universität München, Munich, 80333, Germany
[3]Research Center Jülich, Institute of Bio- and Geosciences 3: Agrosphere, Jülich, 52428, Germany
[4]Department of Earth, Ocean, and Atmospheric Sciences, University of British Columbia, Vancouver, BC, V6T 1Z4, Canada
[5]Present address: Department of Earth Sciences, University of Hong Kong, Hong Kong, China
[6]Research Center for Limnology, Indonesian Institute of Sciences (LIPI), Cibinong-Bogor, Indonesia
[7]Department of Earth Sciences, University of Geneva, Geneva, 1205, Switzerland
[8]Department of Earth, Environmental, and Planetary Sciences, Brown University, 13 Providence, RI, 02912, USA
[9]Faculty of Mining and Petroleum Engineering, Institut Teknologi Bandung, 15 Bandung, 50132, Indonesia
[10]Institute of Geological Sciences and Oeschger Centre for Climate Change Research, University of Bern, CH-3012, Bern, Switzerland
[11]Department of Microbiology and Immunology, University of British Columbia, Vancouver, BC, V6T 1Z3, Canada
*A full list of authors appears at the end of the paper

*Correspondence to*: Aurèle Vuillemin (a.vuillemin@lrz.uni-muenchen.de)

**Abstract.** Ferruginous lacustrine systems, such as Lake Towuti, Indonesia, are characterized by a specific type of phosphorus cycling in which hydrous ferric iron (oxyhydr)oxides trap and precipitate phosphorus to the sediment, which reduces its bioavailability in the water column and thereby restricts primary production. The oceans were also ferruginous during the Archean, so understanding the dynamics of phosphorus in modern-day ferruginous analogues may shed light on the marine biogeochemical cycling that dominated much of Earth's history. Here we report the presence of large crystals (>5 mm) and nodules (>5 cm) of vivianite – a ferrous iron phosphate – in sediment cores from Lake Towuti, and address the processes of vivianite formation, phosphorus retention by iron and the related mineral transformations during early diagenesis in ferruginous sediments.

Core scan imaging together with analyses of bulk sediment and pore water geochemistry document a 30 m long interval consisting of sideritic and non-sideritic clayey beds and diatomaceous oozes containing vivianites. High-resolution imaging of vivianite revealed continuous growth of crystals from tabular to rosette habits that eventually form large (up to 7 cm) vivianite nodules in the sediment. Mineral inclusions like millerite and siderite reflect diagenetic mineral formation antecedent to the one of vivianite that is related to microbial reduction of iron and sulfate. Together with the pore water profiles, these data suggest that the precipitation of millerite, siderite, and vivianite in soft ferruginous sediments stems from

the progressive consumption of dissolved terminal electron acceptors and the typical evolution of pore water geochemistry during diagenesis. Based on solute concentrations and modeled mineral saturation indices, we inferred vivianite formation to initiate around 20 m depth in the sediment. Negative $\delta^{56}Fe$ values of vivianite indicated incorporation of kinetically fractionated light $Fe^{2+}$ into the crystals, likely derived from active reduction and dissolution of ferric oxides and transient
ferrous phases during early diagenesis. The size and growth history of the nodules indicate that, after formation, continued growth of vivianite crystals constitutes a sink for P during burial, resulting in long-term P sequestration in ferruginous sediment.

## 1 Introduction

In the lacustrine realm, phosphorus (P) is often the limiting nutrient for primary production (Compton et al., 2000). Its
supply to primary producers in the euphotic zone depends on external fluxes (Manning et al., 1999; Zegeye et al., 2012) and internal recycling as a result of organic matter (OM) mineralization in both the water column and underlying sediments (Katsev et al., 2006; Hupfer and Lewandowski, 2008). Removal of P through burial in sediments depends partly on sorption to iron oxides (Wilson et al., 2010), and because iron oxides tend to dissolve under reducing conditions and long-term anoxia, phosphate burial is sensitive to the oxygenation state of the water column and water-sediment interface (Sapota et al.,
2006; Rothe et al., 2015). In environments with high sulfate ($SO_4^{2-}$) concentrations and sufficient labile OM, microbial $SO_4^{2-}$ reduction usually leads to the formation of sulfides and eventually iron sulfides, that decrease Fe-recycling and the formation of Fe (oxyhydr)oxides in the upper oxygenated sediments, and this in turn decreases the extent to which P is retained in the sediment (Roden and Edmonds, 1997). Formation of iron phosphate minerals such as vivianite (i.e. $Fe_3(PO_4)_2 \cdot 8H_2O$) in response to the accumulation of sedimentary Fe and P (Gächter et al., 1988; Wilson et al., 2010; Rothe et al., 2016) is a
process that, in contrast, can contribute to long-term P retention in the sediment, particularly in ferruginous (anoxic, non-sulfidic) environments (Gächter and Müller, 2003). Although anoxia is commonly thought to promote P release from sediments and its recycling back to the photic zone of the water column, the high ferrous iron concentrations that can develop in ferruginous environments may promote the formation of iron phosphate minerals, thereby restricting P recycling and bioavailability.
Vivianite is a common phosphate mineral in lacustrine systems (Vuillemin et al., 2013; Rothe et al., 2015). It regularly occurs in organic-rich sediments, often in close association with macroscopic organic remains, and when production of sulfide is low (Rothe et al., 2016). Although it is common in eutrophic lakes, presumably due to high P concentrations, its occurrence in ferruginous, oligotrophic lakes, which may be similar to the Archean oceans, is poorly known. In such systems, high Fe concentrations should catalyze vivianite formation, yet low P concentrations may preclude its formation.
Besides P concentrations, low content and reactivity of OM may also narrow rates of Fe reduction and thereby preclude vivianite formation due to limited release of $Fe^{2+}$ to pore water (Lenstra et al., 2018).

As reported from laboratory studies, vivianite nucleation is possible under relatively high concentrations of $Fe^{2+}$ and orthophosphate (solubility product: Ksp = $10^{-36}$) at pH between 6 and 9 (Glasauer et al., 2003; Rothe et al., 2014; Sánchez-Román et al., 2015). It thus forms as a secondary mineral product in response to iron reduction when P concentrations are sufficiently high (Fredrickson et al., 1998; Zachara et al., 1998; O'Loughlin et al., 2013). Despite these requirements, vivianite formation is not restricted to any specific lake trophic state or salinity range and has been shown to form under a broad range of bottom water redox conditions in freshwater to brackish environments (Egger et al., 2015; Dijkstra et al., 2016). However, high salinities and substantial burial of OM can promote microbial reduction of $SO_4^{2-}$ and sulfide production, which tend to restrict the formation of vivianite in the sediment (Lenstra et al., 2018).

In the humid tropics, deep and intense chemical weathering of bedrock often leads to the formation of thick laterite soils residually enriched in iron (oxyhydr)oxides that promote P scavenging (Lemos et al., 2007). Erosion of these soils and subsequent delivery to lakes promotes P deposition and retention in lake sediments (Fagel et al., 2005; Sapota et al., 2006; Rothe et al., 2014). One of such environment is the ancient Malili Lake System, Sulawesi, Indonesia (Lehmusluoto et al., 1995; Haffner et al., 2001), whose catchment is dominated by ultramafic bedrock overlain by thick lateritic soils (Golightly et al., 2010; Morlock et al., 2019). The Malili Lakes are presently characterized by a dearth of $SO_4^{2-}$ (Crowe et al., 2004; Vuillemin et al., 2016) and low biomass (Bramburger et al., 2008), likely because fluxes of iron (oxyhydr)oxides from surrounding soils scavenge P in the soils, rivers and lake surface waters (Crowe et al., 2008; Katsev et al., 2010; Zegeye et al., 2012). In Lake Towuti, environmental and sedimentary processes, such as weathering intensity (Russell et al., 2014; Morlock et al., 2019), lake mixing and bottom water oxygenation (Costa et al., 2015), and fluctuations in lake level and deltaic sedimentation (Vogel et al., 2015; Hasberg et al., 2019) have changed through time and altered the abundance of reactive ferric iron, and potentially P, in the water column and sediment. However, P dynamics have not been intensively studied in this lake.

On the short term, P retention in lake sediments mainly depends on the oxygenation of the water column (Reed et al., 2016), with depletion of the reducible iron pool under oxygen-poor conditions resulting in the release of accumulated P from the sediment (Katsev et al., 2006). For instance in 600 m deep permanently stratified Lake Matano (Crowe et al., 2008), microbial reduction of iron, which takes place below the modern-day oxycline, leads to partial release of adsorbed P into the bottom water and its accumulation over time (Crowe et al., 2008). Like Lake Matano, Lake Towuti, the largest of the Malili Lakes, presently displays an extreme scarcity of $SO_4^{2-}$ (<20 μM), nitrate/nitrite (<5 μM), phosphate ($PO_4^{3-}$ <5 μM) and oxygen depletion below ~130 m water depth (Vuillemin et al., 2016). Evidence for a complete overturn of the water column is absent in recent years, but sediment data indicate that periods of complete overturn and bottom water oxygenation may have occurred in the geological past (Costa et al., 2015). In an oxygenated water column, hydrous ferric oxides could reach the water-sediment interface and prevent P and Fe release from the sediment (Shaffer, 1986; Katsev et al., 2006). In contrast, anoxia should favor the release of P and Fe from surface sediments and pore waters back to the anoxic bottom water, which would fundamentally change the lake's biology. Even though the lake is presently stratified, $PO_4^{3-}$ in the modern anoxic lake is extraordinarily low, implying a sink for P that is stable under anoxic non-sulfidic conditions, like vivianite. Sediment drill

core data, furthermore, indicate that Lake Towuti has undergone large changes in primary productivity through time, suggesting very different P biogeochemical cycling in the past, possibly linked to dynamics in sediment P mineralogy.

The Malili Lakes, including Lake Towuti, thus represent a relevant setting in which to explore the distribution and characteristics of vivianite formed under Fe-rich and fluctuating redox conditions. From May to July 2015, the Towuti

Drilling Project (TDP) recovered more than 1000 m of sediment drill core from three sites in Lake Towuti, including a 113-m-long core dedicated to geomicrobiological studies at site TDP-TOW15-1A (Russell et al., 2016; Friese et al., 2017). The discovery of sedimentary beds containing large vivianite crystals in this core prompted the present study investigating the distribution and characteristics of vivianite and the modes of vivianite formation in response to environmental variability, P sorption processes and early diagenesis of iron phases.

**2 Methods**

**2.1 Study site and drilling operations**

Lake Towuti (2.5°S, 121°E) is a 200 m deep lake that is part of the Malili Lake System (Fig. 1), a chain of five interconnected tectonic lakes seated in ophiolitic rocks covered with thick lateritic soils on Sulawesi Island, Indonesia (Lehmusluoto et al., 1995; Haffner et al., 2001). The Mahalona River, which is Lake Towuti's main inflow to the north,

connects to the upstream Lakes Mahalona and Matano, while the Larona River constitutes Lake Towuti's only outflow to the west (Vogel et al., 2015). Lake Towuti's water column is circumneutral (pH = 8.4 to 7.2), weakly thermally stratified (i.e. 31-28°C) and presently oxygen-depleted below ~130 m depth (Nomosatryo et al., 2013). The water chemistry is dominated by $Mg^{2+}$ and $HCO_3^-$ ions (Lehmusluoto et al., 1995; Haffner et al., 2001).

The TDP coring operations were carried out from May to July 2015 using the International Continental Scientific Drilling

Program (ICDP) Deep Lakes Drilling System (Russell et al., 2016). Hole TDP-TOW15-1A (156 m water depth; hereafter TDP-1A) was drilled in May 2015 with a fluid contamination tracer used to aid geomicrobiological sampling and analysis (Friese et al., 2017). Samples were collected from cores from TDP-1A immediately upon recovery and over 450 samples were subsequently processed in the field for analyses of pore-water chemistry, cell counting and microbial fingerprinting, and organic geochemistry. Pore water was extracted on site from 5-cm-long whole round cores (6.6 cm diameter) that were

cut from the core sections, immediately capped and transferred to an anaerobic chamber flushed with nitrogen to avoid oxidation during sample handling (Friese et al., 2017). Core catchers from TDP-1A were packed into gas-tight aluminum foil bags flushed with nitrogen gas and heat-sealed to keep them under anoxic conditions until mineral extraction. In January 2016, the unsampled remainders of the cores from TDP-1A were split and scanned at the Limnological Research Center, Lacustrine Core Facility (LacCore), University of Minnesota, described macroscopically and microscopically to determine

their stratigraphy and composition (Russell et al., 2016) and then subsampled. Minerals from core catcher sediments were extracted after 3 month of storage, and macroscopically visible vivianite crystals were hand-picked from split TDP-1A cores after 8 months of storage. Except as otherwise noted, all our samples and measurements come primarily from hole TDP-1A.

## 2.2 Total organic carbon, reactive and total iron

Sediment from core catchers from TDP-1A was used to quantify total organic carbon (TOC). Sediment samples were freeze-dried prior to analysis. Carbonate minerals were removed by treating the samples with 20 mL of 5 % HCl at 50º C for 24 hours (Golubev et al., 2009). Following treatment, samples were repeatedly rinsed with deionized water to reach neutral pH,

centrifuged to discard water and freeze-dried. We tested this treatment using 200 mg of technical grade siderite ($FeCO_3$) to evaluate dissolution. Results showed that 85 % of the siderite is efficiently dissolved within the first 2 hours of treatment and 95 to 100 % after 24 hours (Supplementary Fig. S1). About 8 to 10 mg of homogenized decarbonated samples were measured using an elemental analyzer (EuroVector, EuroEA). Combustion was done in an excess of oxygen at 1040 °C. TOC concentrations were calculated from the yield of $CO_2$ after sample combustion in the elemental analyzer. Analytical

precision of the method is ±3 % ($1\sigma$) of the yield of $CO_2$. TOC was recalculated to the content of the whole sample and results are presented in dry mass weight % (wt %).

For reactive and total Fe sequential extraction, we processed 200 mg of sediment according to Poulton and Canfield (2005). The highly reactive Fe pool is defined as the sum of carbonate-associated Fe (acetate extractable Fe), hydrous Fe (oxyhydr)oxides including ferrihydrite and lepidocrocite (0.5 N HCl extractable Fe), ferric (oxyhydr)oxides including

hematite and goethite (dithionite extractable Fe), and magnetite (oxalate extractable Fe). These reagents do not extract the Fe present in pyrite (Henkel et al., 2016). The non-reactive Fe pool is defined as Fe contained in silicate minerals after removal of reactive phases (near boiling 6N HCl extractable Fe) (Bauer et al., 2020). Total Fe was obtained by summing up the highly reactive Fe pools and the non-reactive Fe contained in silicate minerals. Our protocol could dissolve >92% of the Fe from the PACS-2 international reference standard, ensuring high Fe yield from the samples. All Fe concentration

measurements were performed using a Varian AA875 Flame Atomic Absorption Spectrophotometer (Varian, Palo Alto, USA). Precision on triplicate measurements was 1.2 % and our limit of detection was 1500 μg g$^{-1}$ (0.15 wt % or ~10 μmol cm$^{-3}$).

## 2.3 Pore water geochemistry

After transfer of the whole round cores to the anaerobic chamber, pore water within the upper ten meters was extracted

using Rhizon Pore Water Samplers (Rhizosphere research products, Dolderstraat, Netherlands), directly inserted into the soft sediment. Below 10 m depth, we removed the more compact sediment samples from their liner and scraped off all potentially contaminated rims with a sterile scalpel. The remaining sediment was transferred into an IODP-style titanium pore water extraction cylinder (Mannheim et al., 1966) and placed on a 2-column bench top laboratory hydraulic press (Carver Inc., Wabash, USA). Pore water from both shallow and deep sediments was filtered through a sterile 0.2 μm syringe

filter and collected in a glass syringe pre-flushed with nitrogen. For anion analysis, 1 mL of pore water was transferred to a screw neck glass vial (VWR International, USA) and stored at 4°C until analysis.

Dissolved ferrous and ferric iron concentrations were measured in the field via spectrophotometry (Stookey, 1970). Directly after pore water retrieval, we aliquoted 1 mL of pore water sample to 1.6 mL Rotilabo single-use cells (Carl Roth, Karlsruhe, Germany) and stabilized dissolved $Fe^{2+}$ by adding 100 µL of Ferrozine Iron Reagent (Sigma-Aldrich Chemie Munich, Germany). Absorbance of the colored solution was measured at 562 nm with a DR 3900 spectrophotometer (Hach,

Düsseldorf, Germany). To determine pore water total Fe concentrations, 150 µL of hydroxylamine hydrochloride were added to 800 µL of the previous mixture, left to react 10 min to reduce all dissolved $Fe^{3+}$, stabilized by adding 50 µL ammonium acetate and absorbance of the solution measured a second time (Viollier et al., 2000). Pore water total Fe concentrations were found to be the same as $Fe^{2+}$ concentrations, and thus $Fe^{3+}$ is absent in pore water. Detection limit of the method is 0.25 µM. Concentrations of $PO_4^{3-}$ in pore water were measured by spectrophotometry. We aliquoted 0.5 mL pore water to 1.5 mL

disposable cuvettes (Brand Gmbh, Germany) and added 80 µL color reagent consisting of ammonium molybdate containing ascorbic acid and antimony (Murphy and Riley, 1962). Absorbance was measured at 882 nm with a DR 3900 spectrophotometer (Hach, Düsseldorf, Germany). Detection limit of the method is 0.05 µM. $Mn^{2+}$ concentrations were analyzed spectrophotometrically as previously published (Jones et al., 2011 and 2015), following the formaldoxime method (Brewer and Spencer, 1971). Pore water $Ca^{2+}$, $Mg^{2+}$ and $SO_4^{2-}$ concentrations were analyzed by normal and suppressed ion

chromatography as previously described (Vuillemin et al., 2016). Based on a respective signal-to-noise ratio of 3 and 10, detection and quantification limits of the method calibrated on a multi-element standard are 8.3 and 38.5 µM for $Ca^{2+}$, 9.6 and 44.6 µM for $Mg^{2+}$, and 2.0 and 8.4 µM for $SO_4^{2-}$. All samples were measured in triplicates, with reproducibility better than 5 %.

The pH was measured with a portable pH meter (Thermo Scientific Orion, Star A321) calibrated at pH 4, 7 and 10,

respectively. We homogenized 2 mL of sediment in 2 mL of deionized water and measured the supernatant after 2 min, which is the method commonly used to measure pH in organic-rich soil samples. We followed the published method no. 9045B from Black (1973) and calibrated our results based on the Standard Reference Material Catalog (Seward, 1986). We acknowledge that measuring the pH directly in the pore water using microsensors would provide more accurate in situ results (Reimers et al., 1996). Total alkalinity was measured via colorimetric titration on samples of Rhizon extracted and

hydraulically squeezed pore water. Dissolved inorganic carbon (DIC) concentrations were calculated by solving the carbonate system using the pH and alkalinity profiles and borehole temperatures (Jenkins and Moore, 1977). The complete pore water dataset, inclusive of all major cations and anions (Vuillemin et al., 2019a), was used to calculate mineral saturation indices based on pH, alkalinity, pore water concentrations and borehole temperatures, using the PHREEQC v.3 software (Parkhurst and Appelo, 2013).

**2.4 Vivianite identification and crystal separation**

The drill cores from TDP-1A remaining after field sampling of whole core rounds were split at LacCore, University of Minnesota, USA. Split core halves were imaged at a resolution of 10 pixels/mm (~254 dpi) using a Geotek Geoscan-III with

line-scan CCD cameras with fluorescent lights and polarizing filters to reveal core stratigraphy (Russell et al., 2016). Macroscopically visible vivianite crystals were hand-picked from split TDP-1A cores from 5 distinct horizons located between 20 and 50 m sediment depth. Additional mineral separates of vivianite were obtained from core catchers. We mixed 50 mL of sediment with deionized water in a beaker and sonicated the slurry to homogenize and break up clay aggregates. The slurry was then separated with an initial settling time of ~2 min and removal of the supernatant. We separated the magnetic from the non-magnetic fraction in the settled dense fraction by placing a neodymium magnet below the beaker and rinsing out the non-magnetic mineral fraction with deionized water, followed by drying with acetone. Minerals observed under a stereo microscope (Nikon SMZ800) included siderite, vivianite, millerite (i.e. NiS) and detrital pyroxene (i.e. $Ca(Mg, Fe)Si_2O_6$). Pyrite (i.e. $FeS_2$) was not observed. Vivianite crystals, which were identified in the interval from 20 to 50 m sediment depth, were hand-picked under the stereo microscope for further analyses.

## 2.5 X-ray powder diffraction

X-ray diffraction (XRD) patterns were obtained for one concentrated extract of powdered vivianite as well as for 6 samples of freeze-dried bulk sediment from different depths (i.e. 6.3, 12.4, 23.4, 52.7, 66.5, and 82.6 m), using a PANalytical Empyrean X-ray diffractometer (Eindhoven, The Netherlands), operating with a theta-theta goniometer at 40 kV and 40 mA and a PIXcel 3D detector. $CuK\alpha$ radiation was used with a step size from 4.6 to 85° 2Θ and a count time of 1 min per step. The software packages AXS DIFFRACplus EVA and AXS Topas v. 4.2 (both Bruker) were used to identify minerals and select peak references from the mineralogical database.

## 2.6 Field emission scanning electron microscopy

Isolated crystals of vivianite were mounted on 12.7 mm-diameter aluminum stubs with double-sided conductive carbon tape. An entire vivianite crystal was also embedded in epoxy and the stub cut in axial section. An ultra-thin coating of carbon was deposited on the samples by high-vacuum sputter coating using a Leica MED 020 BAL-TEC metallizer. Imaging was carried out using an Ultra 55 Plus Schottky-type field emission scanning electron microscope. This microscope is equipped with an X-ray energy-dispersive (EDX) system and a Thermo Fisher Scientific silicon-drift detector (SDD UltraDry) for elemental analysis. Operating parameters were set at an acceleration voltage of 20 kV, working distance of 12.5 mm for secondary electron and back-scattering electron images, a 120 mm wide aperture, a silicon-drift detector take-off angle of 35°, and acquisition time of 30 s at a reduced count rate and dead time as needed for point analyses. Calculation of particle chemistry was performed by applying the procedure of the Noran System Seven software based on the standardless matrix correction methods ZAF (i.e. atomic number, absorption, fluorescence) and φ(ρz). Quantitative analyses of all detectable elements were normalized to 100 % atomic weight displayed as oxides. The detection limit for EDX ranges between 0.1 and 1 wt %.

## 2.7 Transmission electron microscopy

Preparation of electron-transparent vivianite samples was done with a FEI FIB200TEM focused ion beam (FIB) device. A TEM-ready foil with final dimensions of $15 \times 10 \times 0.1$ μm was cut directly from the carbon-coated polished section using a gallium ion beam under high-vacuum conditions and placed on a carbon film on top of a copper grid. Carbon-coating to prevent charging of the TEM sample was not applied (Wirth, 2009). The FIB-cut TEM foil was surveyed and analyzed using a FEI Tecnai $G^2$ F20 X-Twin transmission electron microscope. The microscope is equipped with an EDAX ultra-thin window EDX system, a Fishione high-angle annular dark-field (HAADF) detector, and a Gatan imaging filter. Operating conditions were set to an acceleration voltage of 200 kV, using normal imaging mode for bright-field and dark-field imaging and scanning transmission electron microscopy mode for HAADF imaging and analytical electron microscopy. All HAADF images were acquired with a camera length of either 75 or 330 mm. The short camera length (75 mm) allows to image Z contrast, whereas the long camera length (330 mm) allows simultaneous imaging of Z contrast and diffraction contrast. Bright-field images were digitally recorded. Semi-quantitative compositional spectra on both crystalline and amorphous phases were obtained from EDX spectrometer within 60-120 s live time. Beam size in scanning transmission electron mode was 1 to 2 nm and applied across the preselected areas during data acquisition. Structural information on crystalline phases was obtained from selected area electron diffraction patterns recorded on image plates for high precision.

## 2.8 Fe isotope analysis

After density separation, vivianite crystals were hand-picked under the stereo microscope and the isolated crystals processed for Fe isotope analyses at the HELGES lab, GFZ Potsdam (von Blanckenburg et al., 2016); however, the presence of some minor inclusions of siderite, silicates and oxides within vivianite crystals could not be ruled out. To avoid dissolution of silicates and oxides, about 5 mg of sample powder was leached with 2M $HNO_3$ for 24 hours at room temperature (von Blanckenburg et al., 2008). Complete dissolution of vivianite and a few solid residual particles were observed. After centrifugation, supernatants (dissolved vivianite and trace siderite) were evaporated in PFA vials on a hot plate at 110°C, then heated in closed vials at 150 °C with $H_2O_2/HNO_3$ and aqua regia to remove all OM. Procedure blanks and reference materials (USGS COQ-1 carbonatite rock, BHVO-2 basalt rock, HanFe pure Fe solution) were processed along with samples for quality control. After evaporation, samples were re-dissolved in 6M HCl and an aliquot of ~100 μg Fe was passed through chromatographic columns (DOWEX AG-X8 resin) to purify Fe from other matrix elements (Schoenberg and von Blanckenburg, 2005). Purity and quantitative recovery of Fe was verified by inductively coupled plasma - optical emission spectrometry (ICP-OES, Varian 720ES) and found to be better than 98 %. Cr and Ni were efficiently separated from Fe, thus eliminating spectral interferences of $^{54}$Cr on $^{54}$Fe and $^{58}$Ni on $^{58}$Fe. Blanks of all procedures were measured by quadrupole ICP-MS (Thermo iCAP-Qc) and contained <10 ng Fe, thus contributing to <0.01 % of processed Fe samples (~100 μg), and are therefore considered negligible.

Prior to isotope analysis, samples were dissolved in 0.3 M $HNO_3$ and diluted to ~5 µg mL$^{-1}$ Fe to match the concentration of the bracketing standard (IRMM-014) within 10 %. Fe isotopic analyses were performed using a Thermo Scientific Neptune multi-collector inductively coupled plasma mass spectrometer (MC-ICP-MS) equipped with a Neptune Plus Jet Interface pump and a quartz-glass spray chamber (double pass cyclon-scott type, Thermo SIS) with a 100 µL min$^{-1}$ self-aspirating PFA nebulizer for sample introduction. Analyses were run in high mass resolution mode (mass resolving power m/Δm (5 %, 95 %) ~9000) to resolve all Fe isotopes from polyatomic interferences (i.e. ArO, ArOH, and ArN). Potential interferences of $^{54}$Cr on $^{54}$Fe and $^{58}$Ni on $^{58}$Fe were monitored at masses $^{52}$Cr and $^{60}$Ni. The sample-standard bracketing method (using IRMM-014 as bracketing standard) was used to correct for instrumental mass bias (Schoenberg and von Blanckenburg, 2005). Isotope ratios ($^{56}$Fe/$^{54}$Fe and $^{57}$Fe/$^{54}$Fe) are reported in the δ-notation in per mil (‰) relative to the international reference material IRMM-014 (e.g., $\delta^{56}$Fe = ($^{56}$Fe/$^{54}$Fe$_{sample}$/$^{56}$Fe/$^{54}$Fe$_{IRMM-04}$ - 1) × 1000). Measurements were repeated between 2 and 8 times in two independent analytical sessions. Results of $\delta^{57}$Fe and $\delta^{56}$Fe all follow mass-dependent isotope fractionation and therefore results are only discussed in terms of $\delta^{56}$Fe with an uncertainty of the method estimated to be ± 0.05 ‰ (2σ) in $\delta^{56}$Fe, as verified during this study by repeated analyses of reference materials and comparison to published reference values (Supplementary Table S1).

## 3 Results

### 3.1 Lithology and core scanning images

The lithology of TDP site 1 is displayed from 0 to 100 m depth (Fig. 2a), ending at the boundary between the predominantly fine-grained lacustrine Unit 1 and the more coarse-grained fluvio-lacustrine Unit 2 (Russell et al., 2016). The upper 100 m of sediment consist largely of alternating dark reddish-grey and brown to dark-green grey lacustrine clay beds (Fig. 2a-b). Turbidites are relatively rare but more common below 50 m and above 25 m depth and two ~5 m beds of diatom ooze occur at ~35 and 45 m depth. We also observed tephras throughout Unit 1. We focus on 5 intervals containing vivianites between 20 and 50 m depth, where sediment types include both red and green clays, diatomaceous oozes and several large tephras (Fig. 2a). Vivianites are mostly found in green clays, often overlain by siderite-rich red clays and, occasionally, turbidites (Supplementary Fig. S3). Diatomaceous oozes are devoid of vivianite and siderite. A more detailed description of the full stratigraphy is published elsewhere (Russell et al., 2016).

### 3.2 Total organic carbon, reactive and total iron

Over the upper 100 m of the sediment sequence at site TDP-1A, TOC values (Fig. 2a) vary between ~6 and 0.2 wt %. The upper 20 m of sediments display concentrations fluctuating between 3.5 and 0.5 wt % with an overall decrease with depth. In the vivianite-bearing interval (20-50 m), values reach maxima of ~3 wt % in the diatomaceous oozes. In the lowermost part

of the record, TOC gradually increases from 1.0 wt % at 50 m depth to 4.0 wt % at 80 m depth, with highest values (~6%) at the bottom of the core just above the peat layer.

Reactive Fe concentrations vary from 7 to 12 wt % within the upper 20 m of the sediment record, fluctuate between 5 and 15 wt % with the vivianite-bearing interval, and remain relatively constant around 15 wt % below. Total Fe concentrations generally fluctuate between 15 and 20 wt % in Unit 1, with the exception of the interval between 50 and 80 m depth where values occasionally reach 25 to 30 wt %. Some of these high values occur within turbidite beds. The lowest values (~7 wt %) are found within the diatomaceous oozes at 35 and 45 m depth, and just above a peat layer at ~100 m depth.

### 3.3 Pore water geochemistry and modeled saturation indices

In the upper 20 m of sediment, pore water DIC concentrations increase gradually from 2 to 6 mM with depth. Values drop to 4 mM at 20 m depth and then increase gradually from 4 to 7 mM down to 100 m depth. Profiles for pore water $Ca^{2+}$ and $Mg^{2+}$ concentrations display similar trends, with maximum values observed within the upper 15 m of sediment followed by a drop of their respective values from 200 and 600 μM to close to zero. Pore water $PO_4^{3-}$ concentrations in the upper 10 m of sediments increase gradually from 0 to 0.62 μM with depth. Between 20 and 50 m depth, values remain low (0.15 to 0.20 μM). Concentrations of pore water $Mn^{2+}$ are initially ~5 μM in the upper meter of the sediment, minimal in the next 10 m of sediment, and then increase to 20 μM down to 20 m depth. With the vivianite bearing-interval, $Mn^{2+}$ concentrations gradually decrease to a minimum at 50 m depth. Concentrations of pore water $SO_4^{2-}$ were often close to our quantification limit (8.4 μM), with concentrations between 10 and 25 μM. Slightly increased concentrations are observed around 75 m depth. Concentrations of pore water $Fe^{2+}$ are highly variable throughout the sedimentary sequence (17-278 μM). Some of the intervals with highest dissolved $Fe^{2+}$ values are found in the uppermost part of the record (1-6 m), from 15-20 m depth just above the vivianite interval, and in the lowermost section of the core (90-100 m). Below 50 m depth, both $Fe^{2+}$ and $PO_4^{3-}$ values generally peak in the vicinity of turbidite layers.

Geochemical modeling of the pore water chemistry indicates supersaturation with respect to siderite at 5 m depth (1.29) and over the entire lower sediment sequence (Table 1). In contrast, vivianite is undersaturated in pore water at 5 m (-0.45) and reaches close to, but remains below saturation (-0.04) in sediment at 10 m depth. Talc/serpentine is supersaturated in shallow sediment and becomes undersaturated with depth, whereas quartz is stable under in situ conditions. XRD spectra further support the presence and stability of specific phases, such as siderite, quartz, and serpentine (Supplementary Fig. S2).

### 3.4 Iron isotopes

Iron isotopes measured on single vivianite crystals (Fig. 2a) display $\delta^{56}Fe$ values of -0.52 ‰ and -0.44 ‰ at 23 m depth, -0.61 ‰ at 36 m depth, and -0.39 ‰ and -0.46 ‰ at 46 m depth (all ±0.05 ‰, 2 σ). We observe the most negative $\delta^{56}Fe$ values in a specimen from the middle of the vivianite-bearing interval. The iron incorporated in the measured vivianite crystals is isotopically lighter in comparison to the global bulk igneous rock reservoir ($\delta^{56}Fe$ = +0.1 ± 0.1 ‰, e.g. Dauphas et

al., 2017 and references therein), which is the value expected for the ultramafic igneous rocks in Lake Towuti's catchment. To the best of our knowledge, there are no existing data on vivianite $\delta^{56}Fe$ in the literature that would allow comparison. As such, Fe isotope fractionation factors remain unknown for vivianite formation. However, previous studies indicate that during Fe redox reactions, the $Fe^{2+}$-bearing phases generally become enriched in the lighter Fe isotopes compared to $Fe^{3+}$-

bearing phases (e.g., Dauphas et al. 2017). Given that vivianite is a $Fe^{2+}$-bearing mineral phase, the isotopically light $\delta^{56}Fe$ values we measured in vivianites from Lake Towuti are consistent with the direction of fractionation occurring during $Fe^{3+}$ reduction. However, dissolution of precursory ferrous phases could also be the source of the $Fe^{2+}$ incorporated in vivianite crystals.

### 3.5 Vivianite detection, SEM imaging, EDX analysis

The XRD pattern of our powdered vivianite extract confirms identification of this mineral, with an excellent match to reference peaks of one synthetic vivianite from the mineral database (Fig. 3c). Larger vivianite concretions were not observed upon inspection of split core surfaces. Vivianite was also not detected in XRD patterns of bulk sediment at 6 different depths. Siderite, quartz (i.e. $SiO_2$) and serpentine (i.e. lizardite: $Mg_3Si_2O_5(OH)_4$) were the main minerals clearly identified based on reference peaks in these 6 samples, whereas vivianite was below the XRD detection limit of 1 %

(Supplementary Fig. S2).

SEM images of single vivianite crystals (Fig. 3a) show that the habit varies from tabular crystals at 23 m depth to rosette at 36 m depth with addition of blades and overall growth at 46 m depth, and the largest crystal (>7 cm) being found at 50 m depth. EDX points of analysis indicate partial substitution of $Fe^{2+}$ by $Mn^{2+}$ in the structure of vivianite crystals from 23 and 36 m depth (up to ~17% Fe substitution), resulting in their overall "manganoan" composition (Fig. 3b). Such compositions

have been previously reported from both freshwater and marine sediments although with variable Mn concentrations (Fagel et al., 2005; Dijkstra et al., 2016). High Fe content outside the stoichiometric range of vivianite indicates the presence of residual oxides within the crystals. SEM images with location of all EDX points of analysis are available as supplement (Supplementary Fig. S4).

### 3.6 EDX elemental mapping, TEM imaging

SEM images of the vivianite thin section in back-scattering electron mode reveal a central tabular crystal and imply growth of subsequent blades with preferential orientation directed towards the sediment surface (Fig. 4a). Close-ups also reveal the presence of mineral inclusions entrapped within the central tabular blade and upper side of the vivianite (Fig. 4a), namely siderite (1), millerite (2) and goethite (3). Siderite appears in the form of aggregated nanocrystals, millerite in a micro-acicular habit forming radiating aggregates, and goethite in irregular sheets whose jagged edges and dissolution features

likely indicate a detrital origin and potentially partial sedimentary dissolution from iron reduction. EDX elemental mapping (Fig. 4b) as well as individual analyses (Fig. 4c) confirms the composition and identity of these inclusions. Increased intensities of Fe and Mn correspond to goethite, S and Ni to millerite, while those of Si and Al indicate the presence of

phyllosilicates inside and between vivianite blades. Ternary diagrams for individual EDX analyses show that millerite incorporates traces of iron, whereas traces of Mn in iron oxides indicates goethite rather than hematite in which Fe substitution by Mn is limited (Singh et al., 2000).

Scanning TEM imaging of Z contrast and diffraction contrast (i.e. 330 mm) show that the vivianite sample from 46.8 m depth has a denser structure than the one from 36.7 m depth. Close-up images reveal the presence of iron oxides (a), illite clays (b) and detrital pyroxene (c), as confirmed by their EDX analyses, which show that vivianite incorporates phases of detrital origin. Fractures in the crystal from 36.7 m depth could be due to its partial oxidation and dehydration (Hanzel et al., 1990), or to immaturity relative to the sample from 46.8 m depth. Finally, the high resolution electron diffraction pattern of the deepest (oldest) vivianite sample from 46.8 m depth demonstrates its well-ordered monoclinic structure (d), whereas the pattern of the one from 36.7 m depth is somewhat kinked.

## 4 Discussion

### 4.1 Early microbial diagenesis and vivianite growth

In aquatic systems and surface sediments, Fe chemistry influences the distribution of dissolved sulfide, the solubility of trace metals, and bioavailability of P and thereby controls their rates of burial (Severmann et al., 2006). The formation of vivianite in sediments often results from small-scale microbially mediated reactions (Rothe et al., 2016), such as reduction of ferric Fe minerals, with partial dissolution and/or precipitation of mineral phases (Rothe et al., 2014; Egger et al., 2015; Tamuntuan et al., 2015; Dijkstra et al., 2016), alongside OM decomposition (Gächter et al., 2003; Hupfer and Lewandowski, 2008). In Fe-rich, $SO_4^{2-}$-poor, oligotrophic settings like Lake Towuti and Lake Matano where $HS^-$ production is minimal (Vuillemin et al., 2016), the first authigenic Fe minerals expected to form via the reduction of ferrihydrite are mixed-valence iron oxides (e.g. green rust, magnetite) instead of sulfides (Crowe et al., 2008; Zegeye et al., 2012; Vuillemin et al., 2019a). In nearby Lake Matano, lake waters contain more than 40 nM Ni and are supersaturated with respect to millerite where sulfide accumulates to low μM concentrations (Crowe et al., 2008). By analogy to Lake Matano, Ni would compete with Fe for sulfide in Lake Towuti and its sediments. Indeed, we observed the presence of diagenetic millerite in the sediment (Fig. 4) which forms due to the preferential reaction of $HS^-$ with dissolved $Ni^{2+}$ instead of $Fe^{2+}$ (Ferris et al., 1987). Since $SO_4^{2-}$ concentrations are mostly below 50 μM in the core (Fig. 2a), potential rates of sulfide production remain very low compared to the Fe delivery flux and $HS^-$ production has a negligible effect on P release from the sediment. Such low $SO_4^{2-}$ concentrations further result in the loss of most sulfate, and increased methanogenesis, within the first upper meter of sediment (Vuillemin et al., 2018). As a result, processes of OM remineralization are predominantly driven by fermentation and methanogenesis (Friese et al., 2018) and DIC steadily increases with depth (Fig. 2a). Thus, in sediments such as Lake Towuti's, siderite is an expected mineral phase, and indeed siderite is abundant in some of Towuti's sediments (Ordoñez et al., 2019; Vuillemin et al., 2019a). This implies that $CO_3^{2-}$ can compete with $PO_4^{3-}$ for available $Fe^{2+}$. Modeled mineral saturation indices confirmed this, as

pore waters are saturated with respect to siderite at 5 m sediment depth and below. In comparison, vivianite remains close to, but slightly below saturation in deep sediments (Table 1).

We observed an apparent progression in vivianite morphology from tabular to rosette with increasing depth down core (Figs. 3a and Supplementary Fig. S4). Vivianite crystals develop radially and vertically during diagenesis, incorporating authigenic phases and detrital silicates within the crystal and between blades (Figs. 4-5, and Supplementary Fig. S5). Authigenic phases (e.g. siderite, millerite) and detrital oxides (e.g. goethite) were trapped within these crystals (Fig. 4a). Millerite is mainly observed in the tabular template, whereas siderite and goethite are found in the upper blades of the crystal (Figs. 4a-4b). The fact that authigenic siderite and millerite are observed within vivianite crystals demonstrates that vivianite forms at a later stage of diagenesis. Vivianite crystals display growth orientation toward the sediment surface, as shown by the development of successive rosettes on site (Supplementary Fig. S5). In pelagic fine sediments, crystals build up to form successive spherules stacked on top of each other reaching sizes of ~4 to 7 cm (Supplementary Figs. S3 and S5).

Concentrations of Fe oxides in Lake Towuti's sediment are high (~ 20 wt %), and iron oxides such as goethite persist in the modern sediment even under full anoxia at the water-sediment interface and below (Sheppard et al., 2019). If $PO_4^{3-}$ could diffuse out of the sediment, the whole-lake Fe, P and oxygen dynamics predict that any P that might escape to the photic zone from deep, anoxic settings can be buried in shallow-water, oxidized sediments. In the deep sediments, pore water $PO_4^{3-}$ concentrations are constantly low in the interval where vivianite crystals are observed (Fig. 2a), suggesting that vivianites could act as a P sink from the pore water to the sediment during diagenesis (Vuillemin et al., 2013, 2014). Dissolved $Fe^{2+}$ concentrations are not particularly low and fluctuate (40-100 μM), suggesting excess Fe relative to P and potentially reflecting dissolution of detrital phases (e.g. ferrihydrite, goethite, hematite) and/or precipitation of authigenic ones (e.g. magnetite, siderite, vivianite). Concentrations of DIC, which is produced during OM degradation, gradually increase with depth (Fig. 2a), suggesting that OM remineralization in shallow sediment is mainly driven by fermentation and methanogenesis rather than microbial Fe reduction (Vuillemin et al., 2018). In the vivianite-bearing intervals, DIC concentrations remain rather constant (4 mM). An explanation for this is that, once pore water $Fe^{3+}$ and $SO_4^{2-}$ concentrations are depleted as a result of microbial reduction within the first meter of sediment, OM remineralization mostly occurs by $CO_2$ reduction and methanogenesis (Friese et al., 2018). The onset of autotrophic methanogenesis is expected to reduce DIC activity in pore water and decrease the rates of $PO_4^{3-}$ uptake by microbes. Moreover, $Ca^{2+}$ and $Mg^{2+}$ concentrations in pore water, which are predicted to control the solubility of $PO_4^{3-}$ in ferruginous systems (Jones et al., 2015), drop around these depths as divalent cations can precipitate during siderite formation (Vuillemin et al., 2019a), suggesting that $PO_4^{3-}$ can then outcompete $CO_3^{2-}$ for available $Fe^{2+}$ and thereby saturate pore water with respect to vivianite (Table 1) Vivianite formation is indeed reported to occur under methanogenic conditions, often initiating below the sulfate-methane transition zone (Reed et al., 2011; Dijkstra et al., 2016), presently located within the upper meter of sediment (Vuillemin et al., 2018). Since pore waters are saturated with respect to siderite at 10 m depth, it is likely that the initial formation of vivianite occurs in parallel with the precipitation of siderite. Inclusions of millerite and siderite within vivianite crystals (Fig. 4) provide additional lines of evidence for microbial processes of pore $Fe^{3+}$ and $SO_4^{2-}$ reduction and DIC production prior to vivianite formation. The

saturation indices modeled for vivianite (Table 1) and downcore profiles of $Mn^{2+}$ and $PO_4^{3-}$ concentrations allow to infer a depth in the sediment at which pore waters initially reached saturation with respect to vivianite (ca. 20 m depth). Such a relationship between dissolved $Mn^{2+}$ and $PO_4^{3-}$ is also consistent with EDX punctual analyses of vivianite crystals that show $Mn^{2+}$ incorporation at an early stage (Fig. 3b, Supplementary Fig. S4).

However, the fact that dissolved $PO_4^{3-}$, $Mn^{2+}$, $Fe^{2+}$ and DIC vary independently also implies a decoupling in their production and consumption rates (e.g. through mineral formation and microbial metabolic consumption), such that they are not simply linked through steady-state OM respiration coupled to Fe reduction.

## 4.2 Past lacustrine conditions promoting vivianite formation during burial

In sulfur-poor, ferruginous settings, vivianite, siderite and magnetite can be formed in the sediments (Postma, 1981)
depending on the local pH, $CO_2$, $PO_4^{3-}$, the amount and reactivity of ferric oxides and OM buried in the sediment (Fredrickson et al., 1998; Glasauer et al., 2003; O'Loughlin et al., 2013). By analogy to hydromorphic soils (Maher et al., 2003; Vodyanitskii and Shoba, 2015), redox conditions at the time of deposition and fluxes of OM and reactive ferric oxides to the sediment would select for siderite or vivianite as the main diagenetic ferrous end-members during burial. In shallow sediment cores spanning the last ~60 kyr, Costa et al. (2015) suggested that elevated Fe concentrations represent time
intervals of enhanced lake mixing. The alternating dark reddish-brown and lighter green/grey beds, in which vivianites were found in the deeper TDP cores (Fig. 2b, Supplementary Fig. S3), also suggest variable oxygenation at the water-sediment interface in the past (Costa et al., 2015; Russell et al., 2016). Some vivianite-bearing beds appear at similar locations in multiple holes, suggesting lake-wide chemical conditions promoted diagenetic growth of vivianite in the sediment during later burial (Supplementary Fig. S4). However, in other cases, vivianites appear sporadically in only one core. Below the
layer in which the biggest vivianite crystal (>7 cm) was found (~50 m sediment depth), we observe increased iron concentrations (>30 %) corresponding to siderite-rich sediments and multiple turbidites (~50 to 60 m; Fig. 2a). Sporadic turbidites could result in discrete layers enriched in siderites, which are sometimes also linked to bottom water oxygenation, due to enhanced deposition of iron oxides and precipitation of $HCO_3^-$ and $Fe^{2+}$ shortly after deposition (Hasberg et al., 2019; Sheppard et al., 2019). Such turbidites may also promote sporadic laterally non-contiguous vivianite formation in one site
(Supplementary Fig. S4). For instance, a large flat-topped vivianite crystal (>4 cm) capped by a turbidite shows how rapid sedimentary processes prevent further growth of this mineral (Supplementary Fig. S5).

Within the vivianite-bearing interval (20-50 m depth), diatomaceous oozes signify relatively high primary productivity, and their corresponding iron concentrations are lowest, which is consistent with the absence of siderite and vivianite therein (Fig. 2a). Below and above this interval, vivianites are rarely present in the sediment, which was confirmed by smear slide
analysis (Russell et al., 2016), and X-ray diffraction (Supplementary Fig. S2). The substantial fossilization of diatoms, with vivianites below and above these sediments, could reflect higher P concentrations in the water column during this time interval compared to present-day levels, pointing either to increased P supply to the basin and/or a change in P recycling. Given the slow sedimentation rates (~0.2 m $kyr^{-1}$) in the upper 10 m of sediments (Russell et al., 2014), it seems likely that

the 20-50 m interval encompasses at least 100 kyrs. During the Last Glacial Maximum (LGM), Towuti's lake level was 15 to 30 m lower than today, possibly resulting in endorheic conditions (Costa et al., 2015; Vogel et al., 2015). By analogy, lake levels may have been lower during preceding glacial phases, at least one of which is likely to be included in the vivianite-bearing interval. While lake shrinkage could affect algal productivity in the remaining waters (Clavero et al., 1993; Schütt, 1998; Bernal-Brooks et al., 2003; Recasens et al., 2015), lower lake levels should promote bottom water oxygenation and burial of Fe-oxides, and thereby suppress P recycling. Tephras, if they bear apatite (i.e. $Ca_5(PO_4)_3$ (OH)) or additional P adsorbed onto their mineral phases, could represent an additional source of P to the lake (Harper et al., 1986; Nanzyo et al., 1997; Ayris and Delmelle, 2012). As P concentrations tend to affect algal phytoplankton productivity as a whole (Zhang and Prepas, 1996; Van der Grinten et al., 2004), high Si concentrations in the lake represent an additional factor promoting the preservation of diatoms over cyanobacteria during sinking and burial. Finally, sediment-starved conditions would also limit P scavenging in the water column. In contrast, increased delivery of detrital iron (oxyhydr)oxides precipitates P to the sediment and initially forms sideritic beds, whereas vivianite formation initiates under slow kinetics deeper in the sediment than for siderite (Postma, 1981) as demonstrated by the incorporation of siderite, millerite and clay minerals in the vivianites (Figs. 4 and 5).

### 4.3 $\delta^{56}Fe$ compositions of vivianites and implications for the Archean rock record

Previous Fe isotope studies of lakes identified either partial oxidation of $Fe^{2+}$ in the water column or microbial iron reduction below the sediment-water interface as the main drivers for Fe mineralization and transformation pathways and isotope fractionation (Teutsch et al., 2009; Song et al., 2011; Liu et al., 2015). Depending on rates of reduction and dissolution (Brantley et al., 2001), dissimilatory microbial reduction of iron releases $Fe^{2+}$ that is up to 2 ‰ lighter than the original substrates (Crosby et al., 2007; Tangalos et al., 2010), therefore iron isotopes are commonly used to trace redox processes related to microbial activity in aquatic sediments (Percak-Dennett et al., 2013; Busigny et al., 2014). During sediment early diagenesis, the preferential dissolution of isotopically light $Fe^{2+}$ leaves behind an increasingly heavier residual Fe pool (Staubwasser et al., 2006), which results in the diffusive accumulation of the light isotopes in the top layer of sediments where they can be adsorbed and incorporated into ferrous Fe phases. The $\delta^{56}Fe$ values reported for pore water $Fe^{2+}$ in lacustrine sediments range between -2 and -1 ‰ and become heavier with depth as authigenic phases form (Song et al. 2011; Percak-Denett et al. 2013). The $\delta^{56}Fe$ values for reduced Fe phases formed in the sediment are highly variable and range from -1.5 to -0.8 ‰ for pyrite (Busigny et al., 2014), -1.6 to 0.3 ‰ for siderite (Johnson et al., 2005) and -0.1 to 0.2 ‰ for magnetite (Percak-Denett et al. 2013).

Compared to the global bulk igneous rock reservoir ($\delta^{56}Fe = +0.1 \pm 0.1$ ‰) and ultramafic rocks (Dauphas et al. 2017) such as those present in Lake Towuti's catchment, the $\delta^{56}Fe$ measured on whole vivianite crystals (-0.61 to -0.39 ‰) reveals incorporation of isotopically fractionated light $Fe^{2+}$ (Fig. 2a), even though traces of detrital iron-bearing minerals and secondary oxides are present within vivianite crystals (Figs 4 and 5). Towuti's Fe mineralogy from source to sink reflects

complex cycling of Fe as iron minerals derived from catchment soils (e.g. goethite, hematite, magnetite) tend to transform into nanocrystalline Fe phases during reductive dissolution in the lake water column and sediment (Tamuntuan et al., 2015; Sheppard et al., 2019). Ferric/ferrous phases precipitating in equilibrium at the oxycline or during mixing events could be abiotically fractionated to 1-2 ‰ heavier/lighter isotope values than the remaining aqueous $Fe^{2+}$ (Bullen et al., 2001; Skulan

et al., 2002; Beard et al., 2010; Wu et al., 2011). After deposition, partitioning of the light Fe isotopes mainly transits through release to pore water (Henkel et al., 2016) implying a succession of mineral transformation and dissolution with internal diagenetic Fe redistribution during burial (Severmann et al., 2006; Scholz et al., 2014). For instance, mixed-valence iron oxides (e.g. green rust), which are authigenic phases that form initially under ferruginous conditions (Zegeye et al., 2012; Vuillemin et al., 2019a) that can react with pore water $HCO_3^-$ and $HPO_4^{2-}$ and can thereby transform into either siderite

or vivianite as the sediment ages (Hansen and Poulsen, 1999; Bocher et al., 2004; Refait et al., 2007; Halevy et al., 2017). Because vivianite formation initiates in the sediment, we infer that vivianite crystals acted as additional traps for the reduced $Fe^{2+}$ released to pore water and that their light $\delta^{56}Fe$ values are consistent with kinetic fractionation related to microbial Fe reduction during early diagenesis, or eventually inherited from post-depositional dissolution of transient ferrous phases. In the latter case, pre-depositional processes of abiotic Fe fractionation related to stratified conditions will require further

investigations.

Whether they relate to microbial reduction in soft ferruginous sediment or past conditions in bottom waters, biotic and abiotic processes that led to the deposition of ancient iron formations remain challenging to interpret on the basis of their Fe mineral assemblages and isotope compositions (Johnson et al., 2013; Posth et al., 2014). Estimates of concentrations of P in deep anoxic waters, as deduced from the Archean rock, record typically range from 40 to 120 μM for Fe and 0.1 to 0.3 μM

for P (Holland, 2006; Konhauser et al., 2007; Jones et al., 2015), which are similar to those presently observed in the pore water of ferruginous analogue Lake Towuti (Fig. 2a). Concerning P diagenesis, it is hypothesized that P availability in the Archean ocean was limited by the lack of terminal electron acceptors and oxidative power used to recycle most of the OM-bound P rather than by scavenging by Fe minerals (Kipp and Stüeken, 2017; Michiels et al., 2017; Herschy et al., 2018). The present $Ca^{2+}$ and $Mg^{2+}$ concentrations in pore water exert apparent control on the precipitation of siderite and/or vivianite

during early diagenesis (Vuillemin et al., 2019a), which is comparable to interpretations of ancient P availability in regards to hydrothermal and continental weathering of mafic rocks (Jones et al., 2015). In this context because secondary P-bearing minerals cannot form if P remains bound to OM, we suggest that the precipitation of millerite, siderite, and vivianite in the sediment constitutes a likely diagenetic sequence stemming from the progressive consumption of dissolved terminal electron acceptors and evolution of pore water geochemistry, along with the related loss of oxidative power during OM

remineralization, with consequent long-term P sequestration .

## 5. Conclusions

Non-steady state conditions likely promoted the sporadic formation of diagenetic vivianites within otherwise siderite-rich sediments during a prolonged interval of ferruginous Lake Towuti's history. Although the source of P is not well constrained, its inputs stimulated diatom productivity and sporadic vivianite formation during diagenesis. Inclusions of millerite, siderite and partially dissolved goethite within vivianite crystals support the assumption that microbial $Fe^{3+}$ and $SO_4^{2-}$ reduction took place prior to vivianite formation. With depth and over time, vivianite crystals grew and changed from tabular to rosette morphologies, including surrounding clays. The corresponding $\delta^{56}Fe$ compositions confirmed that these crystals incorporated microbially fractionated light $Fe^{2+}$ during diagenesis. While these light isotopic signatures may also point to pre-depositional Fe fractionation related to lake stratification and dissolution of transient ferrous phases, the precipitation of millerite, siderite, and vivianite along burial in ferruginous sediment is consistent with the progressive consumption of dissolved terminal electron acceptors and related loss of oxidative power during OM remineralization, which results in long-term sequestration of P as vivianite. Thus, identification of these diagenetic phases could be used to interpret post-depositional processes of microbial reduction, and thereby help constrain early diagenesis timewise.

## Data availability

Present scientific data are archived and publicly available at PANGAEA® Data Publisher for Earth & Environmental Science (Vuillemin et al., 2019b).

## Author contributions

AV designed the study, sampled in the field, extracted vivianite crystals, actively took part in SEM, TEM, and XRD analyses, designed the figures and led the writing of the present manuscript. AF sampled in the field and measured pore water geochemistry. RW operated the TEM. JAS and FvB processed and measured iron isotopes. AMS led XRD analyses. HK led SEM analyses. AL led TOC analyses. KWB processed and measured samples for total iron and pore water iron. SN measured pH in the field. RS measured alkalinity in the field. LO sampled in the field and processed cores at LacCore. DA processed and sampled cores at LacCore. CH fulfilled the research permit procedure. As principal investigators of the Towuti Drilling Project, JMR, SB, HV sampled in the field, processed drill core splitting and imaging at LacCore, processed TOC and Fe concentrations on full cores, and supervised the writing of the present manuscript. SAC sampled in the field, fulfilled the research permit procedure, supervised iron analyses and writing of the present manuscript. JK sampled in the field and at LacCore, supervised geochemical analyses and writing of the present manuscript. The Towuti Drilling Project Science Team actively participated in drilling operations and processing of the cores at LacCore.

**Competing interests**

The authors declare that they have no conflict of interest.

**Acknowledgements**

This research was carried out with partial support from the International Continental Scientific Drilling Program (ICDP), the
U.S. National Science Foundation (NSF), the German Research Foundation (DFG), the Swiss National Science Foundation
(SNSF), PT Vale Indonesia, the Ministry of Research, Education, and Higher Technology of Indonesia (RISTEK), Brown
University, the University of Minnesota, the University of Geneva, GFZ German Research Centre for Geosciences, the
Natural Sciences and Engineering Research Council of Canada (NSERC), and Genome British Columbia. This study was
financially and logistically supported by the DFG ICDP priority program through grants to JK (KA 2293/8-1) and AV (VU
94/1-1), an SNSF grant to AV (P2GEP2_148621) and an NSERC Discovery grant (0487) to SAC.
We thank PT Vale Indonesia, the U.S. Continental Scientific Drilling and Coordination Office, and U.S. National Lacustrine
Core Repository, and DOSECC Exploration Services for logistical support. The research was carried out with permissions
from RISTEK, the Ministry of Trade of the Republic of Indonesia, the Natural Resources Conservation Center (BKSDA),
and the Government of Luwu Timur of Sulawesi. We thank the Director of the Indonesia Research Center for Limnology
(P2L) - Indonesian Institute of Sciences (LIPI), Tri Widiyanto and his staff of P2L-LIPI for their administrative support in
obtaining the Scientific Research Permit. Supervision by the scientific crew of LacCore during core splitting and
subsampling is kindly acknowledged. We also thank Aan Diyanto, Axel J. Kitte, Anja Schreiber, Ilona Schäpan and
Johannes Glodny for their assistance during field sampling, TEM and SEM analyses and mineral extractions, respectively.

**Team members**

Other members of the TDP Science Team are M. Melles, S. Fajar, A. Hafidz, D. Haffner, A. Hasberg, S. Ivory, C. Kelly, J.
King, K. Kirana, M. Morlock, A. Noren, R. O'Grady, J. Stevenson, T. von Rintelen, I. Watkinson, N. Wattrus, S.
Wicaksono, T. Wonik, A. Deino, A. M. Imran, R. Marwoto, L. O. Ngkoimani, L. O. Safiuddin, and G. Tamuntuan.

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

**Figures**

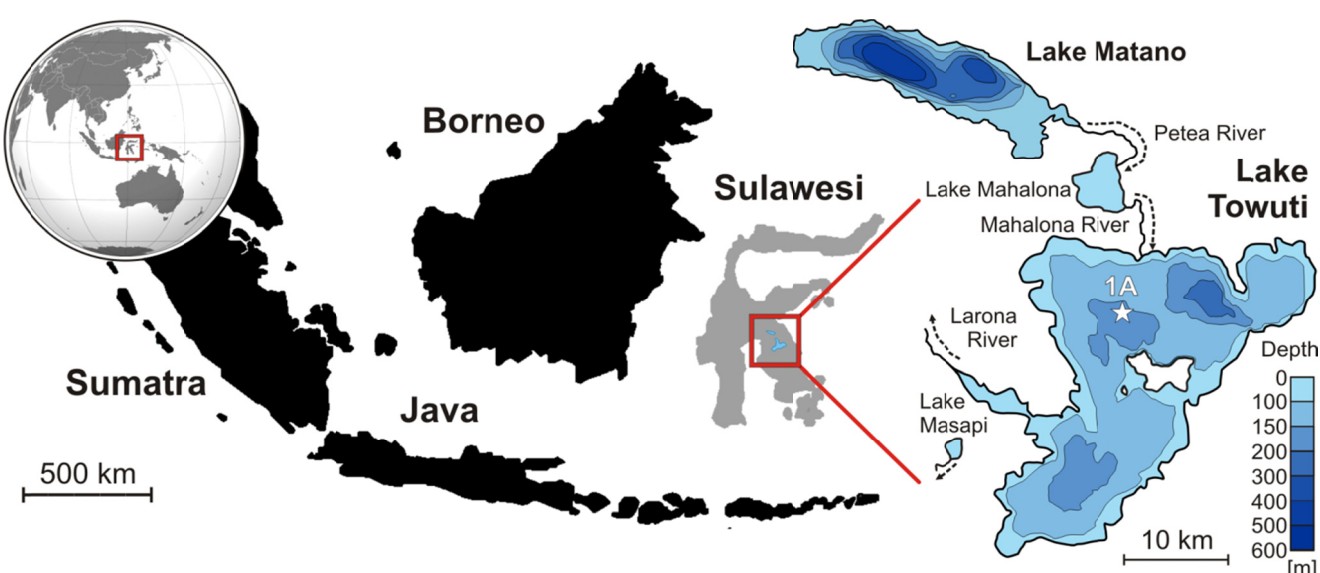

**Figure 1: Location of the Malili Lake System and bathymetric map of Lake Matano and Towuti.** (**Left**) World map displaying the location of Sulawesi Island (red square) with close-up on the Indonesia archipelago and location of the Malili Lake System (red square). (**Right**) Bathymetric map of Lake Matano and Lake Towuti with position of the ICDP drilling site TDP-1A from which hydraulic piston cores were retrieved and sampled for this study.

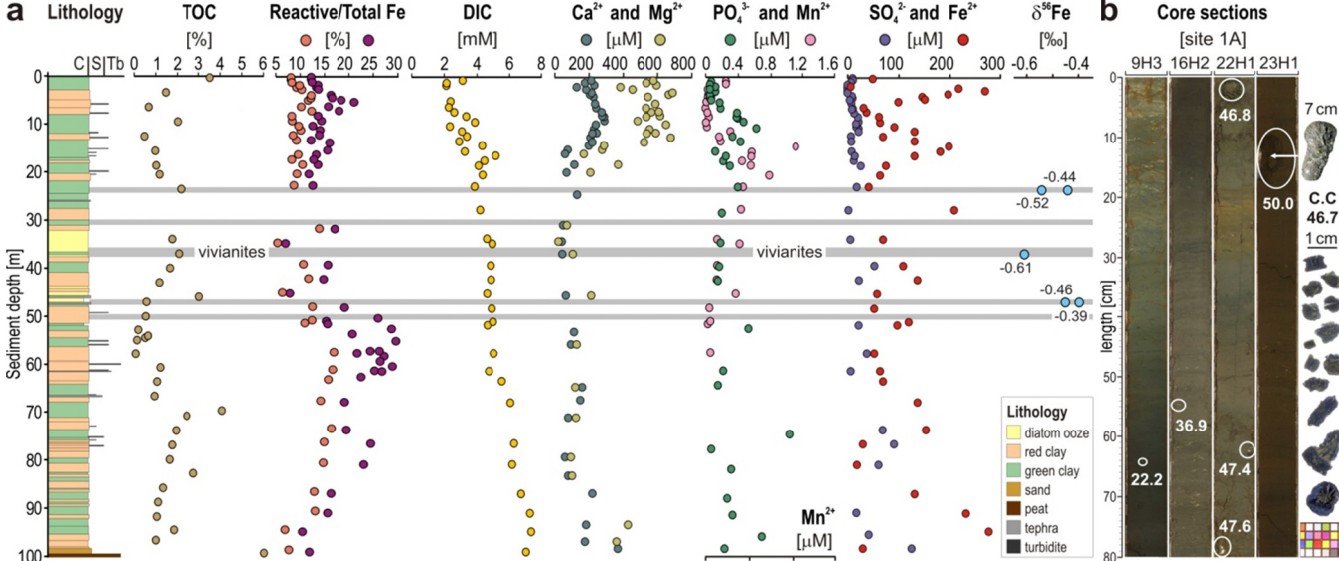

**Figure 2: Stratigraphy of composite site TDP-1, multiple profiles established on sediment core subsamples, and core section images.** (**a**) Stratigraphy of composite site TDP-1 (after Russell et al., 2016) and corresponding grain sizes (C: clay; S: silt; Tb: turbidite); total organic carbon (TOC), reactive and total iron [weight %] in bulk sediment; dissolved inorganic carbon (DIC) [mM], calcium, magnesium, phosphate, manganese, sulfate and ferrous iron concentrations [µM] measured in pore water; and $\delta^{56}$Fe isotopic compositions of vivianite crystals. (**b**) Images of core sections from which vivianite crystals were hand-picked. Crystals on the far right were extracted from core catchers.

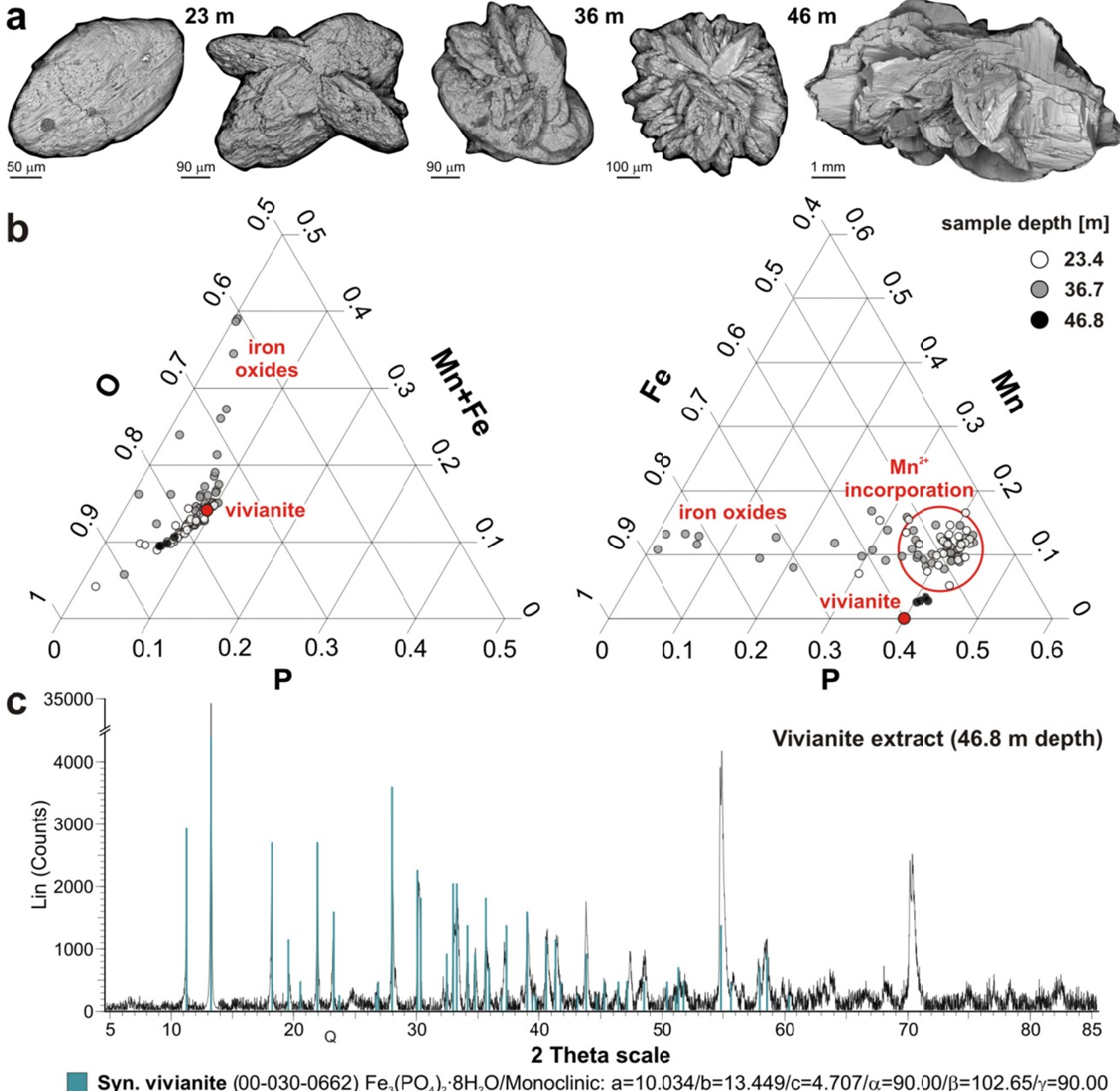

**Figure 3: SEM images of vivianite crystals, ternary diagrams of EDX punctual analyses and XRD spectrum.** (**a**) SEM images show that vivianite crystals grow from a tabular habit to rosette. (**b**) EDX elemental analyses (i.e. O, P, Fe, Mn) of vivianite crystals standardized to 100 % for each ternary diagram. Results indicate incorporation of manganese in the vivianites with the presence of detrital iron oxides. Deepest samples plot closer to stoichiometric vivianite (red dot). (**c**) XRD spectrum of pure vivianite extract from 46.8 m depth with reference peaks of synthetic vivianite (blue bars).

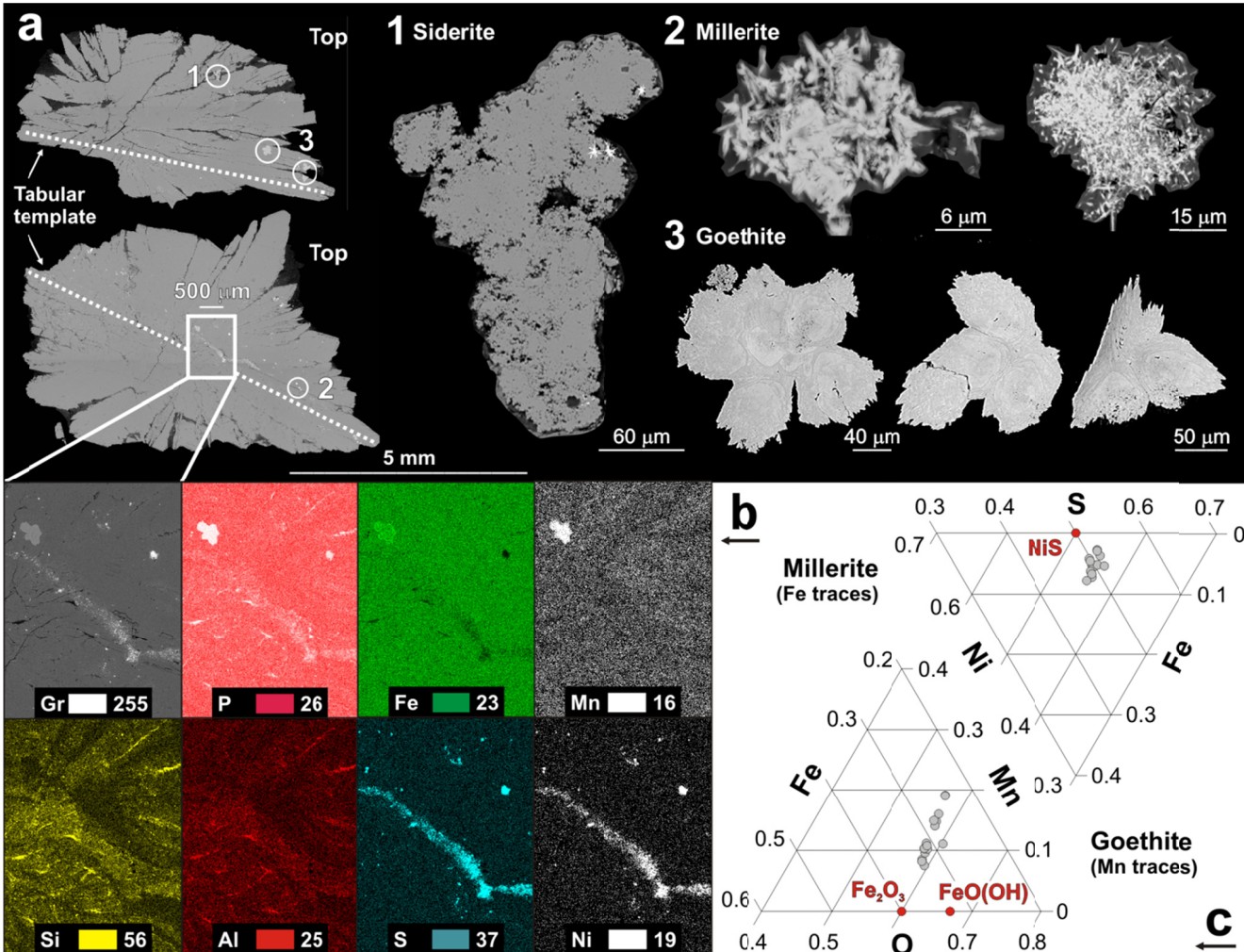

**Figure 4: SEM images of vivianite crystal in axial section, and close-ups to mineral inclusions with EDX mapping and punctual analyses.** (**a**) SEM images of an axial section of a vivianite crystal from 46.8 m depth, with inclusions of siderite (1), millerite (2) and goethite (3). (**b**) EDX elemental mapping of the framed area with relative intensity images for grey levels (Gr), phosphorus (P), iron (Fe), manganese (Mn), silicium (Si), aluminium (Al), sulfur (S) and nickel (Ni). (**c**) Ternary diagrams displaying the elemental composition of millerite and goethite as measured by punctual EDX analyses. Millerite and goethite crystals contain traces of iron and manganese, respectively.

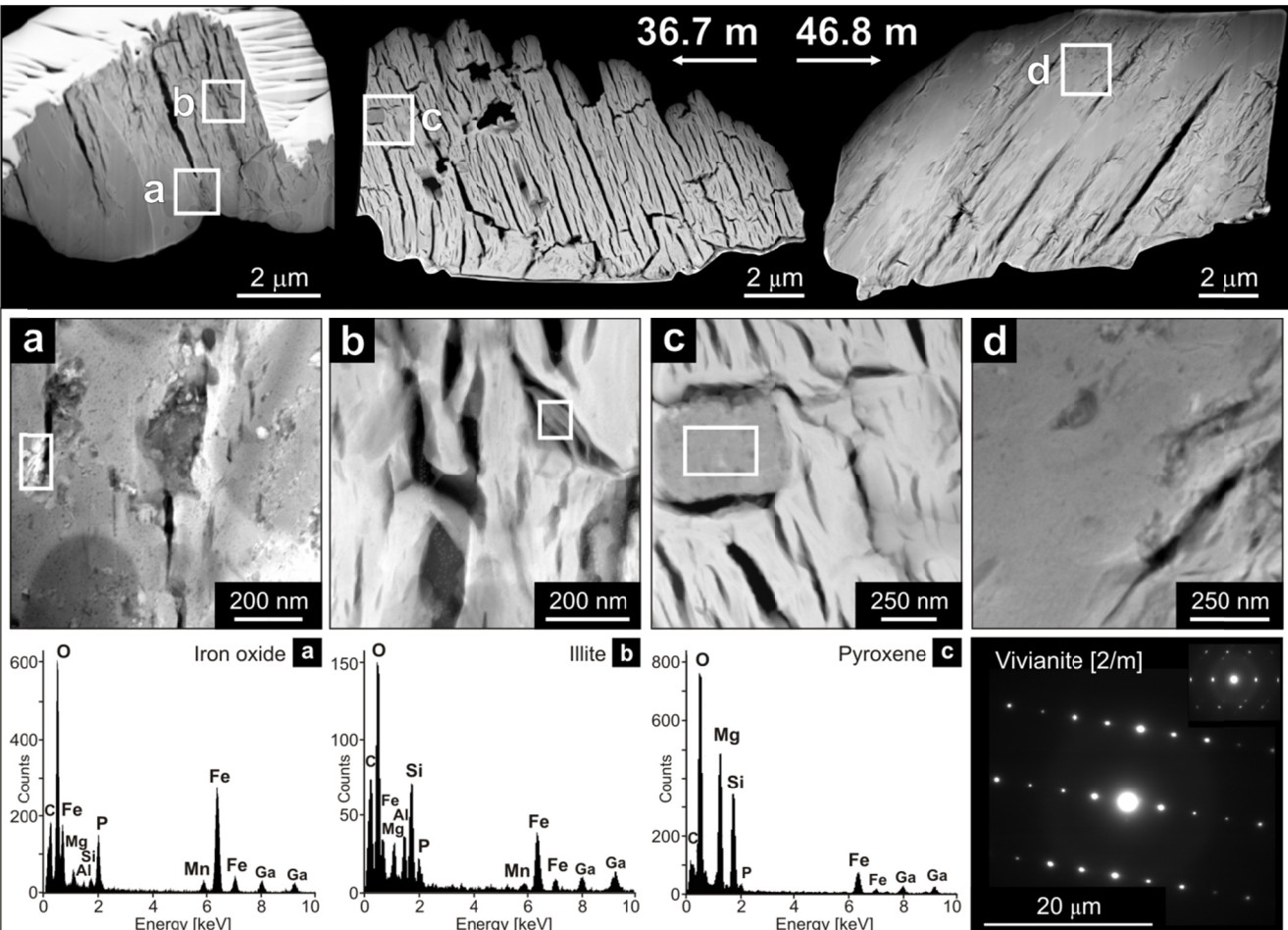

**Figure 5: TEM images of vivianite crystal chunks, and close-ups to detrital inclusions with their EDX analyses and vivianite electron diffraction patterns. (Top)** TEM images (distance: 330 mm) of vivianite crystal chunks from 36.7 and 46.8 m depth, showing that the crystal structure is denser in the deeper sample. (**Middle**) Close-ups of framed areas illustrate the presence of detrital inclusions within vivianite crystals, namely iron oxide (a), illite (b) and pyroxene (c). Image d demonstrates the denser structure of the vivianite crystal from 46.8 m depth. (**Bottom**) EDX spectra for iron oxide (a), illite (b) and pyroxene (c); and high-resolution electron diffraction pattern of the vivianites from 46.8 and 36.7 (insert) m depth providing evidence for an organized and less-organized monoclinic lattice.

**Table**

**Table 1: Modeled saturation indices based on pH, alkalinity, pore water concentrations of major ions and borehole temperatures.**
Siderite appears to be oversaturated throughout the sedimentary sequence, whereas vivianite remains close to, but slightly below saturation with sediment depth.

| 5 m depth | Saturation | 10 m depth | Saturation |
|---|---|---|---|
| talc/serpentine | 1.43 | **siderite** | **1.00** |
| **siderite** | **1.29** | quartz | 0.71 |
| quartz | 0.71 | **vivianite** | **-0.04** |
| **vivianite** | **-0.45** | talc/serpentine | -0.31 |
| calcite | -0.68 | calcite | -0.83 |
| dolomite | -0.77 | aragonite | -0.97 |
| aragonite | -0.82 | dolomite | -1.27 |