# Peer review of "Microbially mediated formation of vivianite during early diagenesis in ferruginous sediments, Lake Towuti, Indonesia."

_Biogeosciences, 2019_

## Referee Comment (RC1) · Anonymous Referee #1 · 4 Dec 2019

General comments The manuscript describes geochemical feature from a long sediment core collected in a deep lake of Indonesia. This lake and a neighbouring lake, Lake Matano, are known for their unique current processes related to iron geochemistry. These lakes have been studied as a proxy for conditions that prevailed in the Archean Ocean. The authors provide new information on iron-phosphorus relationships in this particular context through their study. A good knowledge of phosphorus dynamics in a context of anoxic and non-sulfidic ferruginous water mass can provide interesting knowledge to better understand the phenomena that determined the bioavailability of this nutrient salt during the first episodes of the history of life on Earth. The work presented here focuses mainly on a core more than 100 m long, and more particularly on an interval between 20 and 50 m deep, which contains diatomaceous oozes

and levels rich in vivianite, a ferrous iron phosphate. The authors try to understand how vivianite was formed and under what conditions this mineral traps phosphorus. To do this, they used mineralogical techniques to study the succession of authigenic phases during diagenesis. They also measured dissolved parameters in pore waters and the isotopic signature of iron in vivianite. The issue is of interest to the scientific community, which is seeking to better understand the phosphorus cycle. The discovery and description of the large vivianite crystals is original. While the introduction is well written, some issues related to the results raise questions and the discussion/interpretation of the results is often based on questionable assumptions. This is why I exercise some restraint in validating the publication of the article.

specific comments

Section 2.4 line 27: phosphate was measured by ion chromatography. This method has the disadvantage of having a rather poor detection limit. The authors announce a limit of quantification of 14.3 $\mu$M, which is insufficient here. Commonly used colorimetric methods give at least 20x better detection. The main problem here is that all the concentrations presented in pore water are between 0 and 1 $\mu$M, well below the limit of quantification. Figure 2 shows a PO4 profile with variations between 0 and 1 $\mu$M. There is therefore a problem with these PO4 data, either in the description of the method or in the results; this must be seriously corrected.

line 29 : pH is measured in the supernatant after homogenization of 2 mL sediment in 2 mL deionized water. I have never seen that the pH of pore water can be measured in this way. Is there a reference? The pH is measured here in pore water diluted with deionized water. This cannot give the in-situ pH value. In addition, the authors solve the carbonate system with this questionable pH measurement. Why did the authors not measure the pH in the water used to measure alkalinity?

Section 3.4 line 25: "Given that vivianite is a Fe2+-bearing mineral phase, the isotopically light ïĄd'56Fe values we measured in vivianites from Lake Towuti are consistent

with the direction of fractionation occurring during Fe3+ reduction." This interpretation is based on the principle that Fe(II) is the result of the reduction of Fe(III). But there is no evidence of that. The interpretation given here is indeed the one that would be given for "classical" sediments. But here, the context is very particular and it cannot be excluded that the Fe2+ that produced the vivianite does not come from the dissolution of an initial Fe(II) phase. This point should be discussed.

Discussion section p.11, first paragraph: the first paragraph is not precise enough and is based on too many insinuations. "The ferrous Fe and P released from these reactions may produce vivianite, and/or be consumed through reactions with other dissolved elements such as S." Be more specific. HS- in line8 and H2S in line 13. "Because rates of sulfide production are so low compared to the Fe delivery flux " and "in sediments such as Lake Towuti's, siderite is an expected mineral phase" what are the quantitative elements to assert this?

line 27 "Because of the high concentrations of Fe oxides in Lake Towuti's sediment ($\sim$ 20 wt %), it is very unlikely that much P could escape to the bottom water." This is another unjustified statement. If the ferric phases have their PO4 adsorption sites saturated, PO4 can migrate line 30 "In the deep sediments, pore water PO43- concentrations are constantly low in the interval where vivianite crystals are observed (Fig. 2a), suggesting that vivianites acted as a main P sink during diagenesis". PO4 concentrations are not quantified, given the method used. Are there not the elements to calculate the theoretical concentration of PO4 that would be at thermodynamic equilibrium with vivianite? Such a calculation could strengthen the assertions. p. 12 line 8 "they are (...) rather subject to variety of processes that are variable down core." the conclusion of this paragraph reflects its vague and speculative nature. The processes mentioned should be better described. Methane production suddenly appears without any other explanation. It would be better to specify the presentation.

line 29: "The presence of diatomaceous oozes, with vivianites below and above these sediments, indicates that P concentrations in the water column were much higher during this time interval compared to present-day levels" The authors suggest here that the fossilization of more diatoms suggests that primary production was higher at the time of these deposits and that the increase in primary production was due to an increase in phosphate inputs. If this is the case, it should be detailed as such. But I am not sure that this is necessarily the case, because the exact opposite can be interpreted. Indeed, one could also imagine that an increase in eutrophication due to PO4 input would favour cyanobacteria rather than diatoms, as observed in many lakes. Thus, more diatoms could indicate less PO4. This should be discussed. In addition, the preservation of diatoms may also come from conditions more favourable to their fossilization than other periods.

section 4.3 line 12: "Dissimilatory microbial reduction of iron releases Fe2+ in pore water that is up to 2 ‰ lighter than the original substrates " the fractioning will depend on the rate of reduction. line 21: d56Fe values measured on vivianite are compared to "expected" values for iron oxides. From this point on, all the following in this paragraph is speculative and not supported by data.

Last sentence of the conclusion: "Although crystallization time was not constrained, supply of pore water PO43- and Fe2+ during diagenesis maintained saturation with respect to vivianite and supported the continuous growth of crystals with depth." Here again, this statement is not validated by calculations. The degree of water saturation with respect to vivianite has not been calculated.

Figure and supplementary material are of good quality.

technical corrections

section 2.1, line 10: Describe in more detail the chronology of the operations: date of drilling, extraction of pore water.... the minerals were extracted after more than 6 months of storage. Please specify it.

section 2.2 line 24: the carbonate removal technique to measure the total organic

carbon goes through a washing and centrifugation phase. Doesn't this step cause the loss of organic matter less dense than water?

section 2.4 line17: Dissolved Fe is measured using Ferrozine. Authors have used the method described by Viollier et al., 2000. This study recommends using ascorbic acid to transform all dissolved iron (FeII + FeIII) into FeII. Was ascorbic acid used here?

---

## Referee Comment (RC2) · Anonymous Referee #2 · 7 Jan 2020

In general, it is a nice research from this ICDP project team, and I appreciate the hard work on 100 m of sediment. The topic of vivanite formation in lake sediments is also important and thus fits BG. The authors examined the formation processes in a ferruginous lake, and there are some nice findings. I have, however, some critical comments that should be addressed before this paper can be considered for BG. If the authors cannot address these comments, I would suggest publishing the paper in more specific mineralogical or paleo oriented journal:

Major comments:

The authors should clarify and emphasize their findings. For example, the title is not

informative and strong enough. Just "vivanite formation in sediments" is not enough for justifying publication in BG, and the authors should present an important finding regarding this formation already in the title. Some combination of the sentence in the end of the introduction with results may be used for the title? Please note the same for the abstract and later on. It is important to show what is novel here beyond previous publications of other groups regarding the formation of vivanite (e.g. Slomp, Paytan, Marz, Kasten and more).

It is hard to judge the vivanite formation in response to paleo conditions, because: 1) There is no quantitative investigation of the vivanite (amounts) 2) There is no context of the layers of vivanite to the redox conditions of the porewater. Where are the other porewater profiles? At least $Mn_{2+}$, sulfate, Methane? Is there any SMTZ? Is the vivanite correlate to any of the redox sensitive elements (besides iron. . .)? 3) I'm confused with the paleo interpretation (P. 12). When was the vivanite precipitated in 20-50 m? Thousands years ago at the bottom of the lake or now due to current diagenetic processes? Again more data and discussion (as diffusion rough modeling) are needed to support the first option. It is hard thus to suggest any environmental interpretation without the context of current diagentic processes or quantification of processes. The authors should add the data.

Additional specific comments:

Abstract: The first sentence is not relevant as it refers to ferric iron and phosphate adsorption and not to vivanite. I would write a general sentence instead that states that ferrouginius lakes are important to the phosphorous cycle because of X, Y etc. . ..

Abstract: L 34 is trivial. Add "active" reduction to L 35 to make it also non trivial. It is clear that the redox state is very low in this system to precipitate vivanite, the iron isotopes may suggest its active reduction in this zone. Be careful also with stating it is microbial reduction, as the isotope composition can be light also with abiotic reduction.

Introduction: I do not agree that vivanite is not a studied mineral in sediments, please

correct.

Methods: P. 5: Can the DIC calculated indeed by this approach? How can the authors be sure the alkalinity is mostly carbonatic in this organic rich sediments? Have they measured the carbonate alkalinity or the total alkalinity?

Methods: P. 7: I do not understand how the authors know that they isolate vivanite for the isotope measurement. Please clarify, also in consideration to the fact that diagentic minerals are sometime more reactive to dissolution than detritus ones (look at Henkel's publications).

Discussion: P. 12, see also above. More data is needed and calculations to support precipitation of vivanite at the last glacial.

Discussion P. 13: Again, also abiotic reduction can result in 2 ‰ fractionation.

---

## Referee Comment (RC3) · Anonymous Referee #3 · 7 Jan 2020

This paper describes the formation/presence of vivianite minerals and nodules in lake sediments (Lake Towuti) in Indonesia. They show the presence of large vivianite crystals and nodules in distinct layers is the sediment and discuss how these grow in time. Lake Towuti is a ferruginous lake and the authors argue that this lake could be used as an analogue for the Archean ocean. The paper is very well written and well referenced. The authors used a wide range of techniques to investigate the vivianite minerals found in the sediments. The data is of high quality and is very well presented in clear and structured way. All methods used are clearly described.

General comments

-In my opinion the impact of the paper could increase by adding an implications section at the end of the discussion. Here the authors could present a mass balance for

[Figure]

P and discuss the importance of vivianite in the burial of P in this and other lake sediments. Now it is only briefly mentioned that vivianite might act as the main sink for P at P11.L31.

- The authors mention in the abstract and introduction that Lake Towuti can be used as an analogue for the Archean ocean. However, in the discussion I miss the implications that this study has for the Archean ocean.

- I would like to also see the Fe extraction data for this core. It is mentioned in the method section that Fe extractions were carried out, however, they are now only used to calculate to total Fe present.

Detailed comments:

Introduction

P2.L8: "under anoxia". Here reducing conditions are also important, not only anoxic conditions.

P2.L8: "..phosphate..". Phosphate should be phosphorus (or P) in this case.

P2.L10: This is only the case when there is sufficient organic matter, otherwise there is no formation of sulfide and eventually Fe sulfides.

P2.L12: "Formation of iron phosphate minerals..". Mention that these are reduced iron phosphate minerals.

P2.L22: "In such systems.." Besides the presence of P also the rate/amount of Fe reduction is important in oligotrophic environments. When the organic matter content is low this can lead to limited Fe reduction, low concentrations of porewater Fe and limited formation of vivianite. This has recently been shown in a modeling study for an oligotrophic estuary in the Bothnian Sea (Lenstra et al., 2018; biogeosciences: https://doi.org/10.5194/bg-15-6979-2018)

P2.L27: "..(Egger et al., 2015; Dijkstra et al., 2016)". These studies show viviniate

formation in brackish (not marine) environments. The formation of vivianite, when there is sufficient organic matter, is sensitive to the production of sulfide in the sediment. So at a higher salinity (enhanced sulfide production) the formation of vivianite is expected to be lower. The dependence of vivianite formation on salinity is also discussed in the modeling study I mentioned at P2.L22 (Lenstra et al., 2018; biogeosciences).

P3.L22: "..is stable under anoxic conditions..". Add that it is also important to have non-sulfidic conditions. (anoxic/non sulfidic)

Section 2.3

P5.L2: In these steps you do not extract Fe present in pyrite. I guess this is a very small pool in these environments but to correctly determine the HR Fe pool this should be included or mentioned that this is not included.

P5.L2: How is the non-reactive Fe fraction determined?

Section 2.4

P4.L10: Was this carried out under anoxic conditions?

Section 2.5

P6.L6: "Below and above this interval, vivianites are rarely present in the sediment, which was confirmed by smear slide analysis (Russell et al., 2016) and X-ray diffraction (Supplementary Fig. S2)." This should be moved to the discussion section.

Section 3.2

It would be interesting if you can also show your Fe extraction results in this section. Maybe in the appendix, if you don't want to add an additional figure to the manuscript.

Section 4.1

P11.L24: Is it possible that the orientation of the mineral in the sediment changed during coring? I wonder because the mineral is located very close to the core liner.

P11.L27: Would it be possible to include the solid phase Fe speciation in the paper?

P11.L30: But concentrations of phosphate are generally low not only at places where vivianite is found. I would therefore, based on only the phosphate data, not suggest that vivianite is the main sink of P.

P12.L10: Here and elsewhere Potsma should be Postma

P12.L11: "..depending on the local pH, $CO_2$, $PO_4^{3-}$, and the amount of reactive ferric oxides buried..". Here, also the amount and reactivity of organic matter is important.

Conclusions

P13.L5: I do not understand what partially dissolved iron oxides are.

References

Lenstra, W. K., Egger, M., van Helmond, N. A. G. M., Kritzberg, E., Conley, D. J., and Slomp, C. P.: Large variations in iron input to an oligotrophic Baltic Sea estuary: impact on sedimentary phosphorus burial, Biogeosciences, 15, 6979–6996, https://doi.org/10.5194/bg-15-6979-2018, 2018.

---

## Author Comment (AC1) · 3 Feb 2020

**Reviewer 1**

We thank the reviewer for his/her careful review of our manuscript, which highly helped us improving the new version of the manuscript to be submitted. We have addressed all of the Reviewer's concerns with new additional data, namely:

- Pore water $SO_4^{2-}$, $Ca^{2+}$, $Mg^{2+}$ and $Mn^{2+}$ concentrations were added to Figure 2a and are discussed in the revised manuscript
- Based on geochemical modeling of pore water major ions, we calculated saturation indices for specific minerals. The revised manuscript contains a new table listing saturation indices for diagenetic minerals relevant of the present system (e.g. vivianite, siderite, calcite).
- The description of the methods and results ($SO_4^{2-}$, $PO_4^{3-}$, $Fe^{2+}$, $Mn^{2+}$, $Ca^{2+}$, $Mg^{2+}$) has been carefully reviewed and corrected.

We did our best to fulfill all the remarks and suggestions brought by the Reviewer. Each comment has been addressed in separate answers, and all the corresponding changes are highlighted in red in the text. Please, find here after our point by point answers to Reviewer 1.

Yours sincerely,

Aurèle Vuillemin

**Specific comments**

**Section 2.4**

**- line 27:** Phosphate was measured by ion chromatography. This method has the disadvantage of having a rather poor detection limit. The authors announce a limit of quantification of 14.3 µM, which is insufficient here. Commonly used colorimetric methods give at least 20x better detection. The main problem here is that all the concentrations presented in pore water are between 0 and 1 µM, well below the limit of quantification. Figure 2 shows a $PO_4$ profile with variations between 0 and 1 µM. There is therefore a problem with these $PO_4$ data, either in the description of the method or in the results; this must be seriously corrected.

*Answer 1:* Thank you for detecting this serious mistake from our side. By inadvertence, we reported the method used for major ion quantification. We indeed measured phosphate concentrations using spectrophotometry. We corrected the corresponding method part as follows:

"Concentrations of $PO_4^{3-}$ in pore water were measured by spectrophotometry. We aliquoted 0.5 mL pore water to 1.5 mL disposable cuvettes (Brand Gmbh, Germany) and added 80 µL color reagent consisting of ammonium molybdate containing ascorbic acid and antimony (Murphy and Riley, 1962). Absorbance was measured at 882 nm with a DR 3900 spectrophotometer (Hach, Düsseldorf, Germany). Detection limit of the method is 0.05 µM."
Additional reference:

- Murphy, J., and Riley, J. P.: A modified single solution method for the determination of phosphate in natural waters Anal. Chim. Acta, 27, 31-36, 1962.

**- line 29:** pH is measured in the supernatant after homogenization of 2 mL sediment in 2 mL deionized water. I have never seen that the pH of pore water can be measured in this way. Is there a reference? The pH is measured here in pore water diluted with deionized water. This cannot give the in-situ pH value. In addition, the authors solve the carbonate system with this questionable pH measurement. Why did the authors not measure the pH in the water used to measure alkalinity?

*Answer 2:* What we presently applied is a method commonly used to measure pH in soil samples. We followed the published method no. 9045B from Black (1973) and calibrated our results based on the Standard Reference Material Catalog (Seward, 1986). This information has been added to the text.

"We homogenized 2 mL of sediment in 2 mL of deionized water and measured the supernatant after 2 min, which is the method commonly used to measure pH in organic-rich soil samples. We followed the published method no. 9045B from Black (1973) and calibrated our results based on the Standard Reference Material Catalog (Seward, 1986)."

Additional references:
- Black, C. A. (Ed.): Methods of Soil Analysis: Test Methods for Evaluating Solid Waste, Physical/Chemical Methods, 9045B Soil and Waste pH, American Society of Agronomy, Madison, USA, 1973.
- Seward, R. W. (Ed.): NBS Standard Reference Material Catalog NBS, Special Publication 260, National Bureau of Standards, Gaithersburg, USA, 1986-1987.

**Section 3.4**

**- line 25:** "Given that vivianite is a $Fe^{2+}$-bearing mineral phase, the isotopically light $\delta^{56}Fe$ values we measured in vivianites from Lake Towuti are consistent with the direction of fractionation occurring during $Fe^{3+}$ reduction."

This interpretation is based on the principle that Fe(II) is the result of the reduction of Fe(III). But there is no evidence of that. The interpretation given here is indeed the one that would be given for "classical" sediments. But here, the context is very particular and it cannot be excluded that the $Fe^{2+}$ that produced the vivianite does not come from the dissolution of an initial Fe(II) phase. This point should be discussed.

*Answer 3:* We modified this sentence in part 3.4 with the following statement:

"…are consistent with the direction of fractionation occurring during $Fe^{3+}$ reduction. However, dissolution of precursory ferrous phases could also be the source of the $Fe^{2+}$ incorporated in vivianite crystals."

We rewrote most of section 4.3 in the discussion in order to address Towuti's Fe mineralogy in terms of source to sink processes. We discuss the dissolution of initial and transient ferric/ferrous phases, the neoformation of minerals in the sediment and the related redistribution of Fe isotopes during reductive

diagenesis. We use new pore water data (i.e. $Mn^{2+}$, $Ca^{2+}$, $Mg^{2+}$ and $SO_4^{2-}$ concentrations) and modeled saturation indices for vivianite and siderite to support our interpretations (see answer no. 13).

**Section 4.1**

**- p.11, first paragraph:** "The ferrous Fe and P released from these reactions may produce vivianite, and/or be consumed through reactions with other dissolved elements, such as S."

This first paragraph is not precise enough and is based on too many insinuations. Be more specific.

*Answer 5:* We replaced this sentence by the following one:

"In aquatic systems and surface sediments, Fe chemistry influences the distribution of dissolved sulfide, the solubility of trace metals, and bioavailability of phosphorus and thereby controls their rates of burial (Severmann et al., 2006)…"

Additional reference:
- Severmann, S., Johnson, C. M., Beard, B. L., and McManus, J.: The effect of early diagenesis on the Fe isotope compositions of porewaters and authigenic minerals in continental margin sediments, Geochim. Cosmochim. Ac., 70, 2006-2022, https://doi.org/10.1016/j.gca.2006.01.007, 2006.

**- line 8 and 13:** $HS^-$ and $H_2S$.

*Answer 6:* $H_2S$ modified to $HS^-$ consistently throughout the text.

**- line 16** "Because rates of sulfide production are so low compared to the Fe delivery flux" and "in sediments such as Lake Towuti's, siderite is an expected mineral phase" What are the quantitative elements to assert this?

*Answer 7:* We implemented pore water $SO_4^{2-}$ concentrations to Figure 2a (here under). With the exception of four data points (i.e. 85, 81, 107, 121 µM), $SO_4^{2-}$ concentrations are systematically below 50 µM and are on average 27 µM over the whole profile, which confirms quantitatively that production of $HS^-$ in pore water is limited.

We also modeled mineral saturation indices based on the complete pore water dataset using the PHREEQC v.3 software, inclusive of all major cations ($Na^+$, $NH_4^+$, $K^+$, $Mg^{2+}$, $Ca^{2+}$, $Si^{4+}$, $Fe^{2+}$, DIC), anions ($Cl^-$, $SO_4^{2-}$, $NO^{3-}$, $PO_4^{3-}$), alkalinity, pH and borehole temperatures. These results show that pore water is oversaturated with respect to siderite (>0) in the entire sediment column, whereas vivianite remains close to, but slightly below saturation (-0.04). We report these modeled indices in a new table (Table 1, here under) and implemented the manuscript accordingly.

Additional reference:

- Parkhurst, D. L., and Appelo, C. A. J. (Eds.): Description of input and examples for PHREEQC version 3: a computer program for speciation, batch-reaction, one-dimensional transport, and inverse geochemical calculations, book 6, chapter A43: U.S. Geological Survey Techniques and Methods, Denver, USA, 2013.

[Figure]

Revised Figure 2

| 5 m depth | Saturation | 10 m depth | Saturation |
|---|---|---|---|
| talc | 1.43 | **siderite** | **1.00** |
| **siderite** | **1.29** | quartz | 0.71 |
| quartz | 0.71 | **vivianite** | **-0.04** |
| **vivianite** | **-0.45** | talc | -0.31 |
| calcite | -0.68 | calcite | -0.83 |
| dolomite | -0.77 | aragonite | -0.97 |
| aragonite | -0.82 | dolomite | -1.27 |

Table 1

**- line 27:** "Because of the high concentrations of Fe oxides in Lake Towuti's sediment (20 wt %), it is very unlikely that much P could escape to the bottom water."

This is another unjustified statement. If the ferric phases have their $PO_4$ adsorption sites saturated, $PO_4$ can migrate.

***Answer 8:*** We rephrased the second part of this sentence as follows:

"Concentrations of Fe oxides in Lake Towuti's sediment are high (~ 20 wt %), and iron oxides such as goethite persist in the modern sediment even under full anoxia at the water-sediment interface and below (Sheppard et al., 2019). If $PO_4^{3-}$ could diffuse out of the sediment,…"

**- line 30:** "In the deep sediments, pore water $PO_4^{3-}$ concentrations are constantly low in the interval where vivianite crystals are observed (Fig. 2a), suggesting that vivianites acted as a main P sink during diagenesis".

$PO_4$ concentrations are not quantified, given the method used. Are there not the elements to calculate the theoretical concentration of $PO_4$ that would be at thermodynamic equilibrium with vivianite? Such a calculation could strengthen the assertions.

*Answer 9:* As mentioned in answer no. 1, we corrected the description of the method. Since $PO_4^{3-}$ concentrations were measured via spectrophotometry, our values can be considered reliable and quantitatively sound. As explained in answer no. 7, we also modeled mineral saturation indices based on the complete pore water dataset, which provide theoretical values on pore water saturation with respect to both siderite and vivianite. This new data is available as Table 1.

**- p. 12, line 18**: "they are…rather subject to variety of processes that are variable down core."

The conclusion of this paragraph reflects its vague and speculative nature. The processes mentioned should be better described. Methane production suddenly appears without any other explanation. It would be better to specify the presentation.

*Answer 10:* We deleted this part of the sentence. We rephrased the sentences discussing methanogenic conditions and the role of $CO^2$ reduction by methanogenic archaea in controlling pore water activity of dissolved $CO_3^{2-}$ ions. We also implemented Figure 2a with pore water $Ca^{2+}$, $Mg^{2+}$ and $Mn^{2+}$ concentrations and discuss their role in controlling $PO_4^{3-}$ solubility and reactivity toward $Fe^{2+}$. The text (p. 12-13, lines 30-9) was modified as follows:

"…In contrast, $Fe^{3+}$ concentrations in pore water are already depleted along the upper meter of sediment, whereas concentrations of DIC, which is produced during OM degradation, gradually increase with depth (Fig. 2a), suggesting that OM remineralization in shallow sediment is mainly driven by fermentation and methanogenesis rather than microbial Fe reduction (Vuillemin et al., 2018). In the vivianite-bearing intervals, DIC concentrations remain rather constant (4 mM). An explanation for this is that, once pore water $Fe^{3+}$ and $SO_4^{2-}$ concentrations are depleted as a result of microbial reduction within the first meter, OM remineralization mostly occurs by $CO_2$ reduction and methanogenesis (Friese et al., 2018). The onset of autotrophic methanogenesis is expected to reduce DIC activity in pore water and to draw down $PO_4^{3-}$ cycling by microbes. Moreover, $Ca^{2+}$ and $Mg^{2+}$ concentrations in pore water, which are predicted to control the solubility of $PO_4^{3-}$ in ferruginous systems (Jones et al., 2015), drop around these depths as divalent cations can precipitate during siderite formation (Vuillemin et al., 2019a), suggesting that $PO_4^{3-}$ can then outcompete $CO_3^{2-}$ for available $Fe^{2+}$ and thereby saturate pore water with respect to vivianite (Table 1). Vivianite formation is indeed reported to occur under methanogenic conditions, often initiating below the sulfate-methane transition zone (Reed et al., 2011; Dijkstra et al., 2016), presently located within the upper meter of sediment (Vuillemin et al., 2018). Inclusions of millerite and siderite within vivianite crystals (Fig. 4) provide additional lines of evidence for microbial processes of pore water $Fe^{3+}$ and $SO_4^{2-}$ reduction with DIC production prior to vivianite formation. The saturation indices modeled for vivianite (Table 1) and downcore profiles of $Mn^{2+}$ and $PO_4^{3-}$ concentrations allow to infer a depth in the sediment at which pore waters initially reached saturation with respect to vivianite (ca. 20 m depth). Such relationship between dissolved $Mn^{2+}$ and

$PO_4^{3-}$ is also consistent with EDX punctual analyses of vivianite crystals that show $Mn^{2+}$ incorporation at an early stage (Fig. 3b, Supplementary Fig. S4)…."

Additional references:

- Friese, A., Kallmeyer, J. Glombitza, C., Vuillemin, A., Simister, R., Nomosatryo, S., Bauer, K., Heuer, V. B., Henny, C., Crowe, S. A., Ariztegui, D., Bijaksana, S., Vogel, H., Melles, M., Russell, J. M., and Wagner, D.: Methanogenesis predominates organic matter remineralization in a ferruginous, non-sulfidic sedimentary environment, EGU General Assembly Conference Abstracts , 20, 7446, 2018.
- Jones, C., Nomosatryo, S., Crowe, S. A., Bjerrum, C. J., and Canfield, D. E.: Iron oxides, divalent cations, silica, and the early earth phosphorus crisis, Geology, 43, 135-138, http://doi.org/10.1130/G36044.1, 2015.

**- line 29:** "The presence of diatomaceous oozes, with vivianites below and above these sediments, indicates that P concentrations in the water column were much higher during this time interval compared to present-day levels"

The authors suggest here that the fossilization of more diatoms suggests that primary production was higher at the time of these deposits and that the increase in primary production was due to an increase in phosphate inputs. If this is the case, it should be detailed as such. But I am not sure that this is necessarily the case, because the exact opposite can be interpreted. Indeed, one could also imagine that an increase in eutrophication due to $PO_4$ input would favor cyanobacteria rather than diatoms, as observed in many lakes. Thus, more diatoms could indicate less $PO_4$. This should be discussed. In addition, the preservation of diatoms may also come from conditions more favorable to their fossilization than other periods.

*Answer 11:* We modified this sentence and added another one to address primary productivity by diatoms and cyanobacteria under increased P concentrations and their relative preservation in the sediment. These two sentences are as follows:

"…The substantial fossilization of diatoms, with vivianites below and above these sediments, could reflect higher P concentrations in the water column during this time interval compared to present-day levels…"

"…As P concentrations tend to affect algal phytoplankton productivity as a whole (Zhang and Prepas, 1996; Van der Grinten et al., 2004), high Si concentrations in the lake represent an additional factor promoting the preservation of diatoms over cyanobacteria during sinking and burial…."

Additional references:

- Van der Grinten, E., Janssen, M., Simis, S. G. H., Barranguet, C., and Admiraal, W.: Phosphate regime structures species composition in cultured phototrophic biofilms, Freshwater Biol., 49, 369-381, https://doi.org/10.1111/j.1365-2427.01189.x, 2004.
- Zhang, Y., and Prepas, E. E.: Regulation of the dominance of planktonic diatoms and cyanobacteria in four eutrophic hardwater lakes by nutrients, water column stability, and temperature, Can. J. Fish. Aquat. Sci., 53, 621-633, https://doi.org/10.1139/f95-205, 1996.

**Section 4.3**

**- line 12:** "Dissimilatory microbial reduction of iron releases $Fe^{2+}$ in pore water that is up to 2 ‰ lighter than the original substrates " the fractioning will depend on the rate of reduction.

*Answer 12:* Agreed. We modified the first part of this paragraph as follows:

"Previous Fe isotope studies of lakes identified either partial oxidation of $Fe^{2+}$ in the water column or microbial iron reduction below the sediment-water interface as the main drivers for Fe pathways and isotope fractionation (Teutsch et al., 2009; Song et al., 2011; Liu et al., 2015). Depending on rates of Fe reduction and dissolution (Brantley et al., 2001), dissimilatory microbial reduction of iron releases $Fe^{2+}$ that is up to 2 ‰ lighter than the original substrates (Crosby et al., 2007; Tangalos et al., 2010), therefore iron isotopes are commonly used to trace redox processes related to microbial activity in aquatic sediments (Percak-Dennett et al., 2013; Busigny et al., 2014)…."

**- line 21:** $\delta^{56}Fe$ values measured on vivianite are compared to "expected" values for iron oxides. From this point on, all the following in this paragraph is speculative and not supported by data.

*Answer 13:* As mentioned in answer no. 3, the second paragraph of section 4.3 was entirely rewritten to present Towuti's Fe mineralogy in terms of source to sink processes. We address the dissolution of initial and transient ferric/ferrous phases, the neoformation of minerals in the sediment and the related redistribution of Fe isotopes during reductive diagenesis. We further refer to previous publications on Towuti's Fe mineralogy (Tamuntuan et al., 2015; Sheppard et al., 2019), and use new pore water data (i.e. $Ca^{2+}$, $Mg^{2+}$ and $SO_4^{2-}$ concentrations) and modeled saturation indices to support our interpretations. The second part of section 4.3 was modified as follows:

[revised manuscript text omitted]

Here again, this statement is not validated by calculations. The degree of water saturation with respect to vivianite has not been calculated.

*Answer 14:* We deleted this sentence. As mentioned in answer no. 7, the revised manuscript now includes a new table of modeled saturation indices (Table 1) showing that pore water reaches conditions that are very close to saturation with respect to vivianite with depth. If pore water $PO_4^{3-}$ was indeed incorporated in vivianite crystals during burial, this $PO_4^{3-}$ cannot be accounted for in the pore water modeling. One can thus only infer a timeframe during which geochemical conditions in pore water were saturated with respect to vivianite. In this context, the downcore profiles for pore water $Mn^{2+}$ and $PO_4^{3-}$ concentrations allow to infer the depth at which pore waters reach initial saturation with respect to vivianite (ca. 20 m depth).

**Technical corrections**

**- Section 2.1, line 10:** Describe in more detail the chronology of the operations, i.e. date of drilling, extraction of pore water.... the minerals were extracted after more than 6 months of storage. Please specify it.

*Answer 15:* For detailed information on field sampling and core processing, the readership can refer to (Russell et al., 2016; Friese et al., 2017). Following the reviewer's suggestion, we implemented part 2.1 as follows:

"The TDP coring operations were carried out from May to July 2015 using the International Continental Scientific Drilling Program (ICDP) Deep Lakes Drilling System (Russell et al., 2016). Hole TDP-TOW15-1A (156 m water depth; hereafter TDP-1A) was drilled in May 2015 with a fluid contamination tracer used to aid geomicrobiological sampling (Friese et al., 2017). Samples were collected from cores from TDP-1A immediately upon recovery and over 450 samples were subsequently processed in the field for analyses of pore-water chemistry, cell counting and microbial fingerprinting, and organic geochemistry. Core catchers from TD-1A were packed into gas-tight aluminum foil bags flushed with nitrogen gas and heat-sealed to keep them under anoxic conditions until mineral extraction. Pore water was extracted on site from 5-cm-long whole round cores (6.6 cm diameter) that were cut from the core sections, immediately capped and transferred to an anaerobic chamber flushed with nitrogen to avoid oxidation during sample handling (Friese et al., 2017). In January 2016, the unsampled remainders of the cores from TDP-1A were split and scanned at the Limnological Research Center, Lacustrine Core Facility (LacCore), University of Minnesota, described macroscopically and microscopically to determine their stratigraphy and composition (Russell et al., 2016) and then subsampled. Minerals from core catcher sediments were extracted after 3 month of storage, and macroscopically visible vivianite crystals were hand-picked from split TDP-1A cores after 8 months of storage. Except as otherwise noted, all our samples and measurements come primarily from hole TDP-1A."

**- Section 2.2, line 24:** The carbonate removal technique to measure the total organic carbon goes through a washing and centrifugation phase. Doesn't this step cause the loss of organic matter less dense than water?

*Answer 16:* This is the standard procedure applied for TOC and bulk organic matter using HCl (5%) on freeze-dried samples. We tested this method on pure siderite to ensure that this carbonate phase dissolves in the same way as calcite. Such sample processing is routine analysis at the Forschungszentrum Jülich.

**- Section 2.4, line 17:** Dissolved Fe is measured using Ferrozine. Authors have used the method described by Viollier et al., 2000. This study recommends using ascorbic acid to transform all dissolved iron (FeII + FeIII) into FeII. Was ascorbic acid used here?

*Answer 17:* In Viollier et al., (2000), the authors mention the addition of hydroxylamine hydrochloride as the reducing agent used to measure pore water $Fe^{3+}$, and not ascorbic acid as stated by the reviewer (which is included in the measurement of $PO_4^{3-}$ concentrations). $Fe^{3+}$ concentrations can be calculated as the difference between total Fe and $Fe^{2+}$. We did process the samples using hydroxylamine hydrochloride and therefore we corrected the method description part accordingly. However, total Fe concentrations measured in pore water were identical to those of $Fe^{2+}$, and thus $Fe^{3+}$ is absent in pore water. We also referenced the initial protocol from Stookey (1970). The method part was revised as follows:

"Dissolved ferrous and ferric iron concentrations were measured in the field via spectrophotometry (Stookey, 1970). Directly after pore water retrieval, we aliquoted 1 mL of pore water sample to 1.6 mL Rotilabo single-use cells (Carl Roth, Karlsruhe, Germany) and stabilized dissolved $Fe^{2+}$ by adding 100 µL of Ferrozine Iron Reagent (Sigma-Aldrich Chemie Munich, Germany). Absorbance of the colored solution was measured at 562 nm with a DR 3900 spectrophotometer (Hach, Düsseldorf, Germany). To determine pore water total Fe concentrations, 150 µL of hydroxylamine hydrochloride were added to 800 µL of the previous mixture, left to react 10 min to reduce all dissolved $Fe^{3+}$, stabilized by adding 50 µL ammonium acetate and absorbance of the solution measured a second time (Viollier et al., 2000). Pore water total Fe concentrations were found to be the same as $Fe^{2+}$ concentrations, and thus $Fe^{3+}$ is absent in pore water. Detection limit of the method is 0.25 µM."

Additional reference:

- Stookey, L. L.: Ferrozine - A new spectrophotometric reagent for iron, Anal. Chem., 42, 779-781, https: doi.org/10.1021/ac60289a016, 1970.

---

## Author Comment (AC2) · 3 Feb 2020

**Reviewer 2**

We thank the reviewer for his/her review of our manuscript, which helped us improving the new version of the manuscript to be submitted. We have addressed all of the Reviewer's concerns with additional data, namely:

- Pore water $SO_4^{2-}$, $Mn^{2+}$, $Ca^{2+}$ and $Mg^{2+}$ concentrations were added to Figure 2a and are discussed in the revised manuscript.
- Downcore profiles for both highly reactive and total iron in bulk sediment are provided in Figure 2a.
- Based on geochemical modeling of pore water major ions, we calculated saturation indices for specific minerals (e.g. vivianite, siderite). These indices are listed in a new table (Table 1) now included in the revised manuscript.

We did our best to fulfill all the remarks and suggestions brought by the Reviewer. Each comment has been addressed in separate answers, and all the corresponding changes are highlighted in red in the text. Please, find here after our point by point answers to Reviewer 2.

Yours sincerely,

Aurèle Vuillemin

**Major comments**

**- Comment 1:** The authors should clarify and emphasize their findings. For example, the authors should present an important finding regarding vivianite formation already in the title. The same applies to the abstract and later on, and what is novel regarding the formation of vivianite should be emphasized here in comparison to previous publications (e.g. Slomp, Paytan, Marz, Kasten).

*Answer 1:* We changed the title to "Microbially mediated formation of vivianite during early diagenesis in ferruginous sediments, Lake Towuti, Indonesia."

We implemented the abstract with the following sentences to emphasize, as expressly demanded, the novelty and implications of our findings:

"…Together with pore water profiles, these suggest that the precipitation of millerite, siderite, and vivianite in soft ferruginous sediments stems from the gradual microbial reduction of pore water electron acceptors. Based on solute concentrations and modeled mineral saturation indices, we inferred vivianite formation to initiate around 20 m depth in the sediment as the likely result of the decrease in microbial activity and associated P recycling in pore water. Negative $\delta^{56}Fe$ values of vivianite indicated incorporation of kinetically fractionated light $Fe^{2+}$ into the crystals, likely derived from active reduction and dissolution of ferric oxides and transient ferrous phases during early diagenesis. The size and growth history of the nodules indicate that, after formation, continued growth of vivianite crystals constitutes a sink for P during burial, resulting in long-term P sequestration in ferruginous sediment."

**- Comment 2:** It is hard to judge the vivianite formation in response to paleoconditions, because: there is no quantitative investigation of the amounts of vivianite; there is a lack of context on pore water redox conditions in the vivianite-bearing layers (e.g. SMTZ redox sensitive elements other than iron). Where are the other pore water profiles (at least $Mn^{2+}$, $SO_4^{2-}$, methane)?

*Answer 2:* We provide additional pore water data, namely downcore profiles for $SO_4^{2-}$, $Ca^{2+}$, $Mg^{2+}$, and $Mn^{2+}$ concentrations. The downcore profile for $Mn^{2+}$ concentrations (revised Figure 2a) covaries with $PO_4^{3-}$ and correlates the increased Mn content measured via EDX on early stage vivianite crystals. This, in addition to the saturation indices modeled for siderite and vivianite (Table 1), allows us to infer the depth at which pore waters reach initial saturation with respect to vivianite (see answer no. 3).

We emphasize in the manuscript the depth inferred for the SMTZ, which, due very low $SO_4^{2-}$ concentrations, is located within the upper 50 cm of sediments. This was already addressed in a previous publication based on gravity cores (Vuillemin et al., 2018, Environ. Microbiol. 20).

Concerning methane concentrations, because a manuscript addressing methane production is presently under review, we refrain from providing this data. Since there is no preprint for this manuscript, we only cite an abstract related to the manuscript (Friese et al., 2018; EGU abstract). To answer the reviewer's demand, we provide at his/her discretion an insight into methane concentrations here under, which confirms that the SMTZ is found in very shallow sediments and that methane concentrations constantly increase with sediment depth. The trend for methane is similar to the one of DIC.

[Figure]

Because vivianites are only observed from 20 to 50 m depth intercalated with diatom oozes, we further addressed paleoproductivity and changes in the sedimentary regime. Still, we consider that we provide clear and sufficient evidence for the microbially mediated formation of vivianite during early diagenesis (see answer no. 9). We would also like to bring to the attention of the Reviewer that analyses for paleoenvironmental reconstructions are ongoing work and that a first manuscript detailing the lithostratigraphy and sedimentological processes is close to being submitted. The age model, which is the

most critical aspect in reconstructing past climate, is still under preparation. In the meanwhile, one has to accept some inferences and avoid speculations.

Additional references:

- Friese, A., Kallmeyer, J. Glombitza, C., Vuillemin, A., Simister, R., Nomosatryo, S., Bauer, K., Heuer, V. B., Henny, C., Crowe, S. A., Ariztegui, D., Bijaksana, S., Vogel, H., Melles, M., Russell, J. M., and Wagner, D.: Methanogenesis predominates organic matter remineralization in a ferruginous, non-sulfidic sedimentary environment, EGU General Assembly Conference Abstracts , 20, 7446, 2018.

**- Comment 3:** The paleointerpretation is confusing (page 12). When was the vivianite precipitated in the 20-50 m interval: Thousands years ago at the bottom of the lake or now due to current diagenetic processes? More data and discussion (e.g. diffusion modeling) are needed to support the first or second option. Otherwise, it is hard to suggest environmental interpretations without putting the current diagenetic processes into context or quantifying inferred past processes.

*Answer 3:* We now provide saturation indices modeled for specific minerals (e.g. vivianite, siderite) in a new table (Table 1, here under). These results show that siderite is the main mineral to reach saturation in the pore water at relatively shallow sediment depths (5 m), and that the saturation index modeled for vivianite increases with depth, but remains slightly below saturation. This, in addition to pore water concentrations (i.e. $Mn^{2+}$, $Ca^{2+}$, $Mg^{2+}$), allows to infer the depth at which pore waters are initially saturated with respect to vivianite, which is about 15 m depth. We consider that the downcore profile for pore water $Mn^{2+}$ and $PO_4^{3-}$ in parallel confirms this and correlate the increased Mn content measured on vivianite crystals via EDX and its incorporation at an early stage of growth.

| 5 m depth | Saturation | 10 m depth | Saturation |
|---|---|---|---|
| talc | 1.43 | **siderite** | **1.00** |
| **siderite** | **1.29** | quartz | 0.71 |
| quartz | 0.71 | **vivianite** | **-0.04** |
| **vivianite** | **-0.45** | talc | -0.31 |
| calcite | -0.68 | calcite | -0.83 |
| dolomite | -0.77 | aragonite | -0.97 |
| aragonite | -0.82 | dolomite | -1.27 |

Table 1

**Specific comments**

**Abstract**

The first sentence is not relevant as it refers to ferric iron and phosphate adsorption and not to vivianite. Rewrite a general sentence that states that ferruginous lakes are important to the phosphorous cycle because of X, Y, etc.

*Answer 4:* We rephrased the first sentence as follows:

"Ferruginous lacustrine systems, such as Lake Towuti, Indonesia, represent specific cases of phosphorus cycling in which hydrous ferric iron (oxyhydr)oxides trap and precipitate phosphorus to the sediment, which reduces its bioavailability in the water column and thereby restricts primary production."

**- line 34** is trivial. Also add "active" reduction on line 35 to make it also non-trivial. It is clear that the redox state is very low in this system to precipitate vivianite, the iron isotopes may suggest its active reduction in this zone. Be careful also with stating it is microbial reduction, as the isotope composition can be light also with abiotic reduction.

*Answer 5:* This sentence and rest of the abstract has been rewritten as follows:

"…Mineral inclusions like millerite and siderite reflect diagenetic mineral formation antecedent to the one of vivianite that is related to microbial reduction of iron and sulfate. Together with pore water profiles, these suggest that the precipitation of millerite, siderite, and vivianite in soft ferruginous sediments stems from the gradual microbial reduction of pore water electron acceptors. Based on solute concentrations and modeled mineral saturation indices, we inferred vivianite formation to initiate around 20 m depth in the sediment as the likely result of the decrease in microbial activity and associated P recycling in pore water. Negative $\delta^{56}$Fe values of vivianite indicated incorporation of kinetically fractionated light $Fe^{2+}$ into the crystals, likely derived from active reduction and dissolution of ferric oxides and transient ferrous phases during early diagenesis. The size and growth history of the nodules indicate that, after formation, continued growth of vivianite crystals constitutes a sink for P during burial, resulting in long-term P sequestration in ferruginous sediment."

**Introduction**

I do not agree that vivianite is not a studied mineral in sediments. Please correct.

*Answer 6:* We removed "…less studied…" from the sentence.

**Methods**

**- page 5:** Can the DIC be calculated indeed by this approach? How can the authors be sure the alkalinity is mostly carbonatic in this organic-rich sediment? Have they measured the carbonate alkalinity or the total alkalinity?

*Answer 7:* We measured total alkalinity along with the pH in pore water. In case, the relationship between total alkalinity and carbonate alkalinity according to Jenkins and Moore (1977) is the following:

Carbonate alkalinity ($CaCO_3$ mg/L) = 100,000 [$K_2$ ($HCO_3^-$)($10^{pH}$)].

We added the corresponding reference to the method part. We also measured pore water $K^+$, $Na^+$, $Mg^{2+}$, $Ca^{2+}$, $SO_4^{2-}$ and $Cl^-$ concentrations among others (here under, for information only) and checked the cation-anion mass balance. Thus, we are confident that the DIC calculated by this approach is consistent.

[Figure]

Additional reference:

- Jenkins, S. R., and Moore, R. C.: A proposed modification to the classical method of calculating alkalinity in natural waters, J. Am. Water Works Ass. 69: 56-60, https:// doi.org/10.1002/j.1551-8833.1977.tb02544.x, 1977.

**- page 7:** I do not understand how the authors know that they isolate vivianite for the isotope measurement. Please clarify, also in consideration to the fact that diagenetic minerals are sometime more reactive to dissolution than detritus ones (refer to Henkel's publications).

*Answer 8:* Fe isotope analyses were carried out on entire vivianite crystals, not on pools obtained via Fe sequential extraction. We hand-picked vivianite crystals under the binocular from the dense mineral extracts. The separation and identification procedure is explained in part 2.4. These crystals were further used for XRD, SEM and TEM analyses, which resulted in the certain and definitive identification of the mineral vivianite. The same isolated vivianite crystals were processed for Fe isotope analyses. We clarified this in the text, for instance in:

- page 7, lines 9-10: "Minerals observed under a stereo microscope (Nikon SMZ800) included siderite, vivianite, millerite (i.e. NiS) and detrital pyroxene (i.e. $Ca(Mg, Fe)Si_2O_6$). Pyrite (i.e. $FeS_2$) was not observed. Vivianite crystals, which were identified in the interval from 20 to 50 m sediment depth, were hand-picked under the stereo microscope for further analyses."

-page 8, lines 17-19: "After density separation, vivianite crystals were hand-picked under the stereo microscope and the isolated crystals processed for Fe isotope analyses at the HELGES lab, GFZ Potsdam (von Blanckenburg et al., 2016); however, the presence of some minor inclusions of siderite, silicates and oxides within vivianite crystals could not be ruled out."

**Discussion**

**- page 12:** see also above. More data is needed and calculations to support precipitation of vivianite at the last glacial.

*Answer 9:* We do not state anywhere that vivianite crystallization initiated during the Last Glacial. We only provide the paleolacustrine context corresponding to the sediments that surround vivianite nodules and how

these sediments resulted in geochemical conditions that were favorable to the formation of vivianite during diagenesis, and thus later burial.

To clarify the link between lacustrine paleoconditions and post-depositional processes leading to the diagenetic formation of vivianite in this specific sedimentary interval, we modified the title and introductory sentences of this subchapter as follows:

"***Past lacustrine conditions promoting vivianite formation during burial***
In sulfur-poor, ferruginous settings, vivianite, siderite and magnetite can be formed in the sediments (Postma, 1981) depending on the local pH, $CO_2$, $PO_4^{3-}$, the amount and reactivity of ferric oxides and OM buried in the sediment (Fredrickson et al., 1998; Glasauer et al., 2003; O'Loughlin et al., 2013). By analogy to hydromorphic soils (Maher et al., 2003; Vodyanitskii and Shoba, 2015), redox conditions at the time of deposition and fluxes of OM and reactive ferric oxides to the sediment would select for siderite or vivianite as the main diagenetic ferrous end-members during burial. ...."

Otherwise, we provide substantial additional pore water data (i.e. $Ca^{2+}$, $Mg^{2+}$, $Mn^{2+}$ and $SO_4^{2-}$ concentrations; see revised Figure 2) along with modeled saturation indices for siderite and vivianite (Table 1) to allow us to infer the initial depth of formation of vivianite (see answer no 3). In regards of these additional data, we have carefully checked and clarified the discussion.

[Figure]

Revised Figure 2

Additional references:
- Maher, B. A., Alekseev, A., and Alekseeva, T.: Magnetic mineralogy of soils across the Russian Steppe: climatic dependence of pedogenic magnetite formation. Palaeogeogr. Palaeocl. 201, 321-341, doi: 10.1016/S0031-0182(03)00618-7, 2003.
- Vodyanitskii, Y. N., and Shoba, S. A.: Ephemeral Fe(II)/Fe(III) layered double hydroxides in hydromorphic soils: A review, Eurasian Soil Sci., 48, 240-249, doi: 10.1134/S10642293150, 2015.

**- page 13:** Again, also abiotic reduction can result in 2 ‰ fractionation.

*__Answer 10:__* We are aware of this fact. However, cases of abiotic reduction of Fe that reach 2 ‰ isotopic fractionation usually occur at oxic-anoxic interfaces (see Bullen et al., 2001; Wiesli et al., 2004; Wu et al., 2012). Pore waters are anoxic and it is unlikely that such abiotic fractionation could occur in the sediment. We provide multiple lines of evidence that vivianite crystals form in anoxic sediments, and not in the water column. Nevertheless, as answered to Reviewer 1, the second paragraph of section 4.3 was entirely rewritten to present Towuti's Fe mineralogy in terms of source to sink processes. We address the dissolution of initial and transient ferric/ferrous phases, the neoformation of minerals in the sediment and the related redistribution of Fe isotopes during reductive diagenesis. This includes potential abiotic fractionation at the water column oxycline. The second paragraph of section 4.3 has been rewritten as follows:

"…Compared to the global bulk igneous rock reservoir ($\delta^{56}$Fe = +0.1 ± 0.1 ‰) and ultramafic rocks (Dauphas et al. 2017) such as those present in Lake Towuti's catchment, the $\delta^{56}$Fe measured on whole vivianite crystals (-0.61 to -0.39 ‰) reveals incorporation of isotopically fractionated light $Fe^{2+}$ (Fig. 2a), even though traces of detrital iron-bearing minerals and secondary oxides are present within vivianite crystals (Figs 4 and 5). Towuti's Fe mineralogy from source to sink reflects complex cycling of Fe as iron minerals derived from catchment soils (e.g. goethite, hematite, magnetite) tend to transform into nanocrystalline Fe phases during reductive dissolution in the lake water column and sediment (Tamuntuan et al., 2015; Sheppard et al., 2019). Ferric/ferrous phases precipitating in equilibrium at the oxycline or during mixing events could be abiotically fractionated to 1-2 ‰ heavier/lighter isotope values than the remaining aqueous $Fe^{2+}$ (Bullen et al., 2001; Skulan et al., 2002; Beard et al., 2010; Wu et al., 2011). After deposition, partitioning of the light Fe isotopes mainly transits through release to pore water (Henkel et al., 2016) implying a succession of mineral transformation and dissolution with internal diagenetic Fe redistribution during burial (Severmann et al., 2006; Scholz et al., 2014). For instance, mixed-valence iron oxides (e.g. green rust), which are authigenic phases that form initially under ferruginous conditions (Zegeye et al., 2012; Vuillemin et al., 2019a) are highly sorbent for pore water $HCO_3^-$ and $HPO_4^{2-}$ and can thereby transform into either siderite or vivianite as the sediment ages (Hansen and Poulsen, 1999; Bocher et al., 2004; Refait et al., 2007; Halevy et al., 2017). Because vivianite formation initiates in the sediment, we infer that vivianite crystals acted as additional traps for the reduced $Fe^{2+}$ released to pore water and that their light $\delta^{56}$Fe values are consistent with kinetic fractionation related to microbial Fe reduction during early diagenesis, or eventually inherited from post-depositional dissolution of transient ferrous phases. In the latter case, pre-depositional processes of abiotic Fe fractionation related to stratified conditions will require further investigations…"

Additional references:
- Bocher, F., Géhin, A., Ruby, C., Ghanbaja, J., Abdelmoula, M., and Génin, J.-M. R.: Coprecipitation of Fe(II–III) hydroxycarbonate green rust stabilized by phosphate adsorption, Solid State Science, 6, 117–124, https://doi.org/10.1016/j.solidstatesciences.2003.10.004, 2004.
- Brantley, S. L., Liermann, L., and Bullen, T. D.: Fractionation of Fe isotopes by soil microbes and organic acids, Geology, 29, 535-538, https://doi.org/10.1130/0091-7613(2001)029<0535:FOFIBS>2.0.CO;2, 2001.
- Bullen, T. D., White, A. F., Childs, C. W., Vivit, D. V., & Schulz M. S. Demonstration of significant abiotic iron isotope fractionation in nature. *Geology* **29**, 699-702 (2001).

- Halevy, I., Alesker, M., Schuster, E. M., Popovitz-Biro, R., and Feldman, Y.: A key role for green rust in the Precambrian oceans and the genesis of iron formations, Nat. Geosci., 10, 135–139, https://doi.org/10.1038/ngeo2878, 2017.
- Hansen, H. C. B., and Poulsen, I. F.: Interaction of synthetic sulphate "green rust" with phosphate and the crystallization of vivianite, Clay. Clay Miner., 47, 312-318, doi: 10.1346/CCMN.1999.0470307, 1999.
- Henkel, S., Kasten, S., Poulton, S. W., and Staubwasser, M.: Determination of the stable iron isotopic composition of sequentially leached iron phases in marine sediments. Chem. Geol. 421, 93-102, https: doi.org/10.1016/j.chem.geo.2015.12.003, 2015.
- Refait, P., Reffass, M., Landoulsi, J., Sabot, R., and Jeannin, M.: Role of phosphate species during the formation and transformation of the Fe(II–III) hydroxycarbonate green rust, Colloid. Surface A., 299, 29–37, doi: 10.1016/j.colsurfa.2006.11.013, 2007.
- Scholz, F., Severmann, S., McManus, J., Noffke, A., Lomnitz, U., and Hensen, C.: On the isotope composition of reactive iron in marine sediments: Redox shuttle versus early diagenesis. Chem. Geol., 389, 48-59, https://doi.org/10.1016/j.chemgeo.2014.09.009, 2014.
- Severmann, S., Johnson, C. M., Beard, B. L., and McManus, J.: The effect of early diagenesis on the Fe isotope compositions of porewaters and authigenic minerals in continental margin sediments, Geochim. Cosmochim. Ac., 70, 2006-2022, https://doi.org/10.1016/j.gca.2006.01.007, 2006.
- Welch, S. A., Beard, B. L., Johnson, C. M., & Braterman, P. S. Kinetic and equilibrium Fe isotope fractionation between aqueous Fe(II) and Fe(III). *Geochim. Cosmochim. Ac.* **67**, 4231-4250 (2003).
- Wiesli, R. A., Beard, B. L., & Johnson, C. M. Experimental determination of Fe isotope fractionation between aqueous Fe(II), siderite and "green rust" in abiotic systems. *Chem. Geol.* **211**, 343-362 (2004).
- Wu, L., Percak-Dennett, E. M., Beard, B. L., Roden, E. E., & Johnson, C. M. Stable iron isotope fractionation between aqueous Fe(II) and model Archean ocean Fe-Si coprecipitates and implications for iron isotope variations in the ancient rock record. *Geochim. Cosmochim. Ac.* **84**, 14-28 (2012).

---

## Author Comment (AC3) · 3 Feb 2020

**Reviewer 3**

We thank the reviewer for his/her positive and careful review of our manuscript, which highly helped us improving the new version of the manuscript to be submitted. We have addressed all of the Reviewer's concerns with additional data, namely:

- Downcore profiles for both highly reactive and total iron in bulk sediment.
- Pore water $SO_4^{2-}$, $Ca^{2+}$, $Mg^{2+}$ and $Mn^{2+}$ concentrations were added to Figure 2a and are discussed in the revised manuscript.
- Based on geochemical modeling of pore water major ions, we calculated saturation indices for specific minerals (e.g. vivianite, siderite). These indices are listed in a new table (Table 1) now included in the revised manuscript.

We did our best to fulfill all the remarks and suggestions brought by the Reviewer. Each comment has been addressed in separate answers, and all the corresponding changes are highlighted in red in the text. Please, find here after our point by point answers to Reviewer 3.

Yours sincerely,
Aurèle Vuillemin

**General comments**

**- Comment 1:** In my opinion, the impact of the paper could increase by adding an implications section at the end of the discussion. Here the authors could present a mass balance for P and discuss the importance of vivianite in the burial of P in this and other lake sediments. Now it is only briefly mentioned that vivianite might act as the main sink for P (page 11, line 31).

*Answer 1:* We do not have sufficient quantitative measurements of P concentrations and vivianite crystals in bulk sediment to accurately model the P mass balance. To satisfy the Reviewer's interest, we provide at his/her discretion preliminary XRF profiles (here under) and compare them with those of pore water. We are reluctant to provide XRF data for publication or discuss them any further since they are not calibrated and are part of another manuscript in preparation.

To address the relevance of vivianite in P burial, we oriented the discussion around saturation indices modeled for vivianite and siderite (Table 1), and show how pore water saturation with respect to vivianite increases with depth and leads to the gradual depletion of dissolved phosphate with burial.

We further discuss P cycling in the present ferruginous analogue and compare pore water concentrations to those interpreted from the Archean rock record. We added a short paragraph at the end of the discussion summing up the implications of P cycling processes in modern ferruginous sediment and use those interpretations from the rock record of Archean marine sediments (see answer no. 2).

[Figure]

Preliminary XRF profiles

| 5 m depth | Saturation | 10 m depth | Saturation |
|-----------|------------|------------|------------|
| talc | 1.43 | **siderite** | **1.00** |
| **siderite** | **1.29** | quartz | 0.71 |
| quartz | 0.71 | **vivianite** | **-0.04** |
| **vivianite** | **-0.45** | talc | -0.31 |
| calcite | -0.68 | calcite | -0.83 |
| dolomite | -0.77 | aragonite | -0.97 |
| aragonite | -0.82 | dolomite | -1.27 |

Table 1

- **Comment 2:** The authors mention in the abstract and introduction that Lake Towuti can be used as an analogue for the Archean ocean. However in the discussion, I miss the implications that this study has for the Archean ocean.

*Answer 2:* We added a paragraph at the end of the discussion summing up the implications of the diagenetic processes observed in modern ferruginous sediment for the Archean oceans as interpreted from the ancient rock record. This paragraph reads as follows:

"Whether they relate to microbial reduction in soft ferruginous sediment or past conditions in bottom waters, biotic and abiotic diagenetic processes remain challenging to constrain in terms of ancient rock record (Johnson et al., 2013). Concentrations estimates for deep anoxic waters interpreted from the Archean rock record typically range from 40 to 120 μM for Fe and 0.1 to 0.3 μM for P (Holland, 2006; Konhauser et al., 2007; Jones et al., 2015), which are similar to those presently observed in the pore water of ferruginous analogue Lake Towuti (Fig. 2a). Concerning P diagenesis, it is hypothesized that P availability in the Archean ocean was limited by the lack of terminal electron acceptors and oxidative power used to recycle most of the OM-bound P rather than by scavenging by Fe minerals (Kipp and Stüeken, 2017; Michiels et al., 2017; Herschy et al., 2018). The present $Ca^{2+}$ and $Mg^{2+}$ concentrations in pore water exert apparent control

on the precipitation of siderite and/or vivianite during early diagenesis (Vuillemin et al., 2019a), which is comparable to interpretations of ancient P availability in regards to hydrothermal and continental weathering of mafic rocks (Jones et al., 2015). In this context because secondary P-bearing minerals cannot form if P remains bound to OM, we suggest that the precipitation of millerite, siderite, and vivianite in the sediment constitute a likely diagenetic sequence stemming from the depletion of pore water electron acceptors and related loss of oxidative power during OM remineralization, with consequent long-term P sequestration."

Additional references:

- Herschy, B., Chang, S. J., Blake, R., Lepland, A., Abbott-Lyon, H., Sampson, J., Atlas, Z., Kee, T. P., and Pasek, M. A.: Archean phosphorus liberation induced by iron redox geochemistry, Nat. Commun., 9, 1346, https://doi.org/10.1038/s41467-018-03835-3, 2018.
- Kipp, M. A., and Stüeken, E. E.: Biomass recycling and Earth's early phosphorus cycle, Sci. Adv., 3, eaao4795, https://doi.org/10.1126/sciadv.aao4795, 2017.
- Konhauser, K. O., LaLonde, S. V., Amskold, L., and Holland, H. D.: Was there really an Archean phosphate crisis?, Science, 315, 1234, https://doi.org/10.1126/science.1136328, 2007.
- Michiels C.C., Darchambeau, F., Roland, F. A., Morana, C., Llirós, M., Garcia-Armisen, T., Thamdrup, B., Borges, A. V., Canfield, D. E., Servais, P., Descy, J.-P., and Crowe, S. A.: Iron-dependent nitrogen cycling in a ferruginous lake and the nutrient status of Proterozoic oceans, Nat. Geosci., 10, 217-221, https://doi.org/10.1038/NGO2886, 2017.

**- Comment 3:** I would like to also see the Fe extraction data for this core. It is mentioned in the method section that Fe extractions were carried out, however they are now only used to calculate the total Fe present.

*Answer 3:* Because sequential Fe extractions constitute the main part of a manuscript presently in preparation, we refrain from providing the complete dataset. To satisfy the reviewer's legitimate comment, we provide downcore profiles for the highly reactive Fe (i.e. the sum of four successive Fe pools) and total Fe (i.e. reactive + non-reactive) pools in the revised Figure 2a.

We clarified the extraction procedure in the method section, part 2.2, and cite the corresponding manuscript available as preprint at https://www.EarthArXiv.org. We also updated Figure 2a (here under) and complemented the results, part 3.2, accordingly.

Additional reference:

- Bauer, K. W., Byrne, J., Kenward, P., Simister, R., Michiels, C., Friese, A., Vuillemin, A., Henny, C., Nomosatryo, S., Kallmeyer, J., Kappler, A., Smit, M., Francois, R., Crowe, S. A.: Magnetite biomineralization in ferruginous waters and early Earth evolution, EarthArXiv Preprint, https://doi.org/10.31223/osf.io/prhuz, 2020.

[Figure]

Revised Figure 2

**Specific comments**

**Introduction**

**- page 2, line 8:** "under anoxia". Here reducing conditions are also important, not only anoxic conditions.

*Answer 4:* Rephrased to "…reducing conditions and long-term anoxia…"

**- line 8:** "..phosphate..". Phosphate should be phosphorus (or P) in this case.

*Answer 5:* Modified accordingly.

**- line 10:** This is only the case when there is sufficient organic matter, otherwise there is no formation of sulfide and eventually Fe sulfides.

*Answer 6:* Modified accordingly to "….high sulfate ($SO_4^{2-}$) concentrations and sufficient labile OM,…leads to the formation of sulfides and eventually iron sulfides that decrease…"

**- line 12:** "Formation of iron phosphate minerals..". Mention that these are reduced iron phosphate minerals.

*Answer 7:* Mentioned accordingly.

**- line 22:** "In such systems.." Besides the presence of P also the rate/amount of Fe reduction is important in oligotrophic environments. When the organic matter content is low this can lead to limited Fe reduction, low concentrations of pore water Fe and limited formation of vivianite. This has recently been shown in a modeling study for an oligotrophic estuary in the Bothnian Sea (Lenstra et al., 2018)

*Answer 8:* We added the following sentence to the text:

"Besides P concentrations, low content and reactivity of OM may also narrow rates of Fe reduction and thereby preclude vivianite formation due to limited release of $Fe^{2+}$ to pore water (Lenstra et al., 2018)."

We added (Lenstra et al., 2018) to the list of references and cite it where appropriate in the manuscript.

**- line 27:** (Egger et al., 2015; Dijkstra et al., 2016) show vivianite formation in brackish (not marine) environments. The formation of vivianite, when there is sufficient organic matter, is sensitive to the production of sulfide in the sediment. So at a higher salinity (enhanced sulfide production) the formation of vivianite is expected to be lower. The dependence of vivianite formation on salinity is also discussed in the modeling study by (Lenstra et al., 2018).

*Answer 9:* Thank you for this correction. We replaced "marine" by "brackish" in the corresponding sentence. We also added one sentence to clarify the influence of salinity and OM content on sulfide production in the sediment:

"However, high salinities and substantial burial of OM can promote microbial reduction of $SO_4^{2-}$ and sulfide production, which tend to restrict the formation of vivianite in the sediment (Lenstra et al., 2018)."

**- page 3, line 22:** "…is stable under anoxic conditions.." Add that it is also important to have non-sulfidic conditions. (anoxic/non sulfidic).

*Answer 10:* Modified accordingly.

**Section 2.3**

**- page 5, line 2:** In these steps, you do not extract Fe present in pyrite. I guess this is a very small pool in these environments but to correctly determine the HR Fe pool this should be included or mentioned that this is not included.

*Answer 11:* We added the following information in the methods, part 2.2.:

"For reactive and total Fe sequential extraction, we processed 200 mg of sediment according to Poulton and Canfield (2005). The highly reactive Fe pool is defined as the sum of carbonate-associated Fe (acetate extractable Fe), hydrous Fe (oxyhydr)oxides including ferrihydrite and lepidocrocite (0.5 N HCl extractable Fe), ferric (oxyhydr)oxides including hematite and goethite (dithionite extractable Fe), and magnetite (oxalate extractable Fe). These reagents do not extract the Fe present in pyrite (Henkel et al., 2016). The non-reactive Fe pool is defined as Fe contained in silicate minerals after removal of reactive phases (near boiling 6N HCl extractable Fe). Total Fe was obtained by summing up the highly reactive Fe pools and the non-reactive Fe contained in silicate minerals (Bauer et al., 2020)."

**- line 2:** How is the non-reactive Fe fraction determined?

***Answer 12:*** The non-reactive Fe pool is defined as the Fe contained in silicate minerals after removal of reactive phases (near boiling 6N HCl extractable Fe). See answer 11.

**Section 2.4**

**- line 13**: Was this carried out under anoxic conditions?

***Answer 13:*** Yes, all pore water extractions were carried out in an anaerobic chamber. This was already mentioned in part 2.1:

"Pore water was extracted on site from 5-cm-long whole round cores (6.6 cm diameter) that were cut from the core sections, immediately capped and transferred to an anaerobic chamber flushed with nitrogen to avoid oxidation during sample handling (Friese et al., 2017)."

We now mention this once more in part 2.3: "After transfer of the whole round cores to the anaerobic chamber, pore water within the upper ten meters was extracted using Rhizon…"

**Section 2.5**

**- page 6, line 6:** "Below and above this interval, vivianites are rarely present in the sediment, which was confirmed by smear slide analysis (Russell et al., 2016) and X-ray diffraction (Supplementary Fig. S2)." This should be moved to the discussion section.

***Answer 14:*** Moved accordingly to the discussion, part 4.2 (page 14, line 10).

**Section 3.2**

It would be interesting if you can also show your Fe extraction results in this section. Maybe in the appendix, if you don't want to add an additional figure to the manuscript.

***Answer 15:*** Please refer to answers no. 3, 11 and 12.

**Section 4.1**

**- page 11, line 24:** Is it possible that the orientation of the mineral in the sediment changed during coring? I wonder because the mineral is located very close to the core liner.

***Answer 16:*** Although sediment disturbance cannot be fully excluded, we observed multiple vivianites in different core sections, with crystal orientation reflecting an upward growth (Supplementary Fig. S5) so that polarity and growth direction can be inferred without difficulty.

**- line 27:** Would it be possible to include the solid phase Fe speciation in the paper?

***Answer 17:*** See answers no. 3, 11, 12 and 15

**- line 30:** But concentrations of phosphate are generally low not only at places where vivianite is found. I would therefore, based on only the phosphate data, not suggest that vivianite is the main sink of P.

*Answer 18:* Rephrased from "…acted as a main P sink…" to "…could act as a P sink from the pore water to the sediment during diagenesis".

**- page 12, line 10:** Here and elsewhere Potsma should be Postma

*Answer 19:* Corrected accordingly.

**- line 11:** "..depending on the local pH, $CO_2$, $PO_4^{3-}$, and the amount of reactive ferric oxides buried..". Here, also the amount and reactivity of organic matter is important.

*Answer 20:* This was added to the sentence accordingly.

**Conclusions**

**- page 13, line 5:** I do not understand what partially dissolved iron oxides are.

*Answer 21:* Rephrased to "…partially dissolved goethite..."

---

## Author Response (AR2)

**To the Editor and Reviewers**

We thank the Editor and the Reviewers for taking the time to review the revised version of our manuscript and help us to clarify certain aspects of the methods and discussion. We have now addressed all the comments posted by the Reviewers. Please find hereafter our point-by-point answers to each of the Reviewers' comments. The corresponding changes in the latest version of the revised manuscript are highlighted in red font. We hope thereby to finalize the peer-review process and meet criteria for publication in Biogeosciences. We remain at your disposition for further information and are looking forward to the outcome of your deliberations. On behalf of all the authors,

Yours sincerely

Aurèle Vuillemin
(first/corresponding author)

**Reviewer 1**

**- Comment 2.** The method used to determine the pH is not commonly used for sediments, and is unlikely to provide accurate insight in the in-situ pH, as indicated by the reviewer. Typically, pH in sediment pore waters is either determined with microelectrodes (e.g. Reimers et al., 1996; GCA) or by inserting a pH probe in the pore water sample. To address the reviewer's comment fully, a comment providing this context could be added e.g.: "We note, however, that direct measurements of pH in the sediment pore water with, for example, microsensors would provide a better indication of the in-situ pH (e.g. Reimers et al. 1996). "

_**Answer 1:**_ We added the following sentence to the revised manuscript, page 6, lines 22-24: "We acknowledge that measuring the pH directly in the pore water using microsensors would provide more accurate in situ results (Reimers et al., 1996)."

Additional reference:
- Reimers, C. E., Ruttenberg, K. C., Canfield, D. E., Christiansen, M. B., and Martin, J. B.: Porewater pH and authigenic phases formed in the uppermost sediments of the Santa Barbara Basin. Geochim. Cosmochim. Ac., 60, 4037-4057, https://doi.org/10.1016/S0016-7037(96)00231-1,1996.

**Reviewer 2**

**- Comment 1.** I'm not convinced the change in title is supported by your manuscript. Of course practically all redox reactions in sediments are microbially-driven. However, no specific evidence for a key role of a specific type of microbes in inducing vivianite formation is presented but this current title could be interpreted as such. This title also could be misunderstood to indicate that you find evidence that the vivianite is formed in microbial cells (which we know is possible). I would suggest keeping the original title.

_**Answer 1:**_ Replaced accordingly.

**Other comments**

**- P1. Line 23.** "represent specific cases" could be replaced by "are characterized by a specific type of"

*Answer 2:* Modified accordingly.

**- P1. Line 35-36.** Text could be changed to "Together with the pore water profiles, these data suggest…."

*Answer 3:* Modified accordingly.

**- P2. Line 1, P16, line 23 and P17, line 3.** Can you clarify what "pore water electron acceptors" refers to, i.e. can you be more specific with respect to the electron acceptors you are referring to? How do you know the electron acceptors are solutes and not solids? Alternatively, you could keep the text more general and refer to the progression of the typical sequence of electron acceptors during diagenesis.

*Answer 4:* Modified in the abstract, page 2, lines 1-2, as follows: "…stems from the progressive consumption of dissolved terminal electron acceptors and the typical evolution of pore water geochemistry during diagenesis."

Modified in the discussion, page 16, lines 26-28, as follows: "…from the progressive consumption of dissolved terminal electron acceptors and evolution of pore water geochemistry along with the related loss of oxidative…"

Modified in the conclusions, page 17, lines 6-7, as follows: "…consistent with the progressive consumption of dissolved terminal electron acceptors…"

**- P2. Line 2-3.** It is not clear how the decrease in microbial activity and P recycling are linked here. Could you be more specific? Alternatively, this text (starting at "as the likely…") could be removed.

*Answer 5:* Deleted accordingly.

**- P12.** Why is it relevant to mention "because Ni is also abundant"? What data does that refer to? I would suggest to omit this from the sentence.

*Answer 6:* Deleted accordingly.

**- P13.** It is not clear what is meant by "drawdown $PO_4^{3-}$ cycling by microbes"

*Answer 7:* Rephrased as follows: "… expected to reduce DIC activity in pore water and decrease the rates of $PO_4^{3-}$ uptake by microbes."

**- P13. Line 23-24.** What about parallel precipitation of siderite and vivianite?

*Answer 8:* We added the following sentence, page 13, lines 31-33, to address this possibility: "Since pore waters are saturated with respect to siderite at 10 m depth, it is likely that the initial formation of vivianite occurs in parallel with the precipitation of siderite."

**- P13, starting with line 14:** why are you referring to $Fe^{3+}$ in the pore water whereas your measurements show that $Fe^{3+}$ is absent from in the pore water (see section 2.3).

*Answer 9:* We deleted the first half of this sentence accordingly.

**- P13. Line 30.** "Such a relationship…"

*Answer 10:* Modified accordingly.

**- P15. Line 13.** What is meant exactly by "Fe pathways"? Can you be more specific?

*Answer 11:* Modified to "Fe mineralization and transformation pathways".

**- P16. Line 4.** "are highly sorbent for" could be replaced by "that can react with pore water"

*Answer 12:* Modified accordingly.

**- P16. Line 12.** It is not clear what is meant by "in terms of ancient rock record". I would suggest rephrasing this.

*Answer 13:* Rephrased as follows: "…abiotic processes that led to the deposition of ancient iron formations remain challenging to interpret on the basis of their Fe mineral assemblages and isotope compositions (Johnson et al., 2013; Posth et al., 2014).

Additional reference:
- Posth, N. R., Canfield, D. E., and Kappler, A.: Biogenic Fe (III) minerals: from formation to diagenesis and preservation in the rock record, Earth-Sci. Rev., 135, 103–121, doi: 10.1016/j.earscirev.2014.03.012, 2014.

**- P16. Line 13.** "Concentrations estimates" could be changed to "Estimates of concentrations of P in deep anoxic waters, as deduced from…"

*Answer 14:* Modified accordingly.

[revised manuscript text omitted]

---

## Author Response (AR3)

**To the Editor**

We thank the Editor for the time spent handling and correcting our manuscript. Please find hereafter our point-by-point answers to her latest comments. The corresponding changes in the revised manuscript (merged to the present pdf) are highlighted in red font. We remain at your disposition for further information. On behalf of all the authors,

Yours sincerely

Aurèle Vuillemin
(first/corresponding author)

**Comments and answers**

**- Comment 1:** In addressing the comment regarding the pH, you added the following text: "We acknowledge that measuring the pH directly in the pore water using microsensors would provide more accurate in situ results (Reimers et al., 1996)." This suggests that the method that you used to measure pH is an accurate method for in situ pH values in sediments and that the microsensors are simply more accurate. You provide no citation for this and I doubt that this is correct (in line with the reviewer's comment). Hence, I would propose to use the formulation that I suggested previously: "We note, however, that direct measurements of pH in the sediment pore water with, for example, microsensors would provide a better indication of the in-situ pH (e.g. Reimers et al. 1996)."

*Answer 1:* As expressly demanded, we replaced the former sentence by the following one: "We note, however, that direct measurements of pH in the sediment pore water with, for example, microsensors would provide a better indication of the in situ pH (e.g. Reimers et al. 1996)."

**- Comment 2 (line 23, page 13):** Pore water $Fe^{3+}$ should be pore water $Fe^{2+}$.

*Answer 2:* Since pore waters are not depleted of $Fe^{2+}$ at this depth (ca. 1 m sediment depth) and that $Fe^{2+}$ cannot be further microbially reduced, we decided to simply delete "pore water $Fe^{3+}$" from the corresponding sentence. The new text reads as follows: "An explanation for this is that, once $SO_4^{2-}$ concentrations are depleted as a result of microbial reduction within the first meter of sediment, OM remineralization…"

**- Comment 3 (line 26, page 13):** In addressing the comment regarding "drawdown $PO_4^{3-}$ cycling by microbes" the following text was added: "decrease the rates of $PO_4^{3-}$ uptake by microbes". It is still not clear what this refers to exactly. I would suggest to either clarify or remove.

[revised manuscript text omitted]